# Regulation of long-range BMP gradients and embryonic polarity by propagation of local calcium-firing activity

Hyung Chul Lee [1,9] ✉, Nidia M. M. Oliveira [1,10], Cato Hastings[1], Peter Baillie-Benson[2], Adam A. Moverley[1,3], Hui-Chun Lu[1,11], Yi Zheng [4,5,6], Elise L. Wilby[7], Timothy T. Weil[7], Karen M. Page [8], Jianping Fu [4], Naomi Moris [2] & Claudio D. Stern [1] ✉

Many amniote vertebrate species including humans can form identical twins from a single embryo, but this only occurs rarely. It has been suggested that the primitive-streak-forming embryonic region emits signals that inhibit streak formation elsewhere but the signals involved, how they are transmitted and how they act has not been elucidated. Here we show that short tracks of calcium firing activity propagate through extraembryonic tissue via gap junctions and prevent ectopic primitive streak formation in chick embryos. Cross-regulation of calcium activity and an inhibitor of primitive streak formation (Bone Morphogenetic Protein, BMP) via NF-κB and NFAT establishes a long-range BMP gradient spanning the embryo. This mechanism explains how embryos of widely different sizes can maintain positional information that determines embryo polarity. We provide evidence for similar mechanisms in two different human embryo models and in *Drosophila*, suggesting an ancient evolutionary origin.

During embryo development, cells assess their position (positional information) by sensing different concentrations of a morphogen gradient; such gradients have been proposed to arise by diffusion across the tissue, decreasing away from their source[1]. However, gradients can reliably be generated by diffusion only over a limited range (<100 cell diameters, or 1 mm)[2–4], which raises the question of how patterning is achieved in larger systems. Different species of embryos show huge variation in size (from a few μm to many mm) at the time when gastrulation starts, when the embryo establishes the orientation of the future embryonic axis. A highly conserved feature in all metazoans is that Bone Morphogenetic Proteins (BMPs), which belong to the TGFβ superfamily, are expressed asymmetrically in early embryos, and act, through Smad1/5/8, as inhibitors for the formation of the blastopore (in anamniotes) or of its amniote equivalent, the primitive streak, through which the mesoderm and endoderm ingress during gastrulation[5]. Other TGFβ family members, notably GDF/VG1 and NODAL, are expressed at the opposite end of the embryo and act, through Smad2/3, to induce the blastopore/primitive streak. Bird and

[1]Department of Cell and Developmental Biology, University College London, Gower Street, London WC1E 6BT, UK. [2]The Francis Crick Institute, 1 Midland Road, London NW1 1AT, UK. [3]Department of Cell and Developmental Biology, Institute for Regenerative Medicine, Perelman School of Medicine, University of Pennsylvania, Philadelphia, PA, USA. [4]Departments of Mechanical Engineering, Biomedical Engineering, and Cell & Developmental Biology, University of Michigan, Ann Arbor, MI, USA. [5]Department of Biomedical and Chemical Engineering, Syracuse University, Syracuse, NY, USA. [6]BioInspired Syracuse Institute for Materials and Living Systems, Syracuse University, Syracuse, NY, USA. [7]Department of Zoology, University of Cambridge, Downing Street, Cambridge CB2 3EJ, UK. [8]Department of Mathematics, University College London, Gower Street, London WC1E 6BT, UK. [9]Present address: School of Biological Sciences and Technology, College of Natural Sciences, Chonnam National University, 77 Yongbong-ro, Gwangju 61186, Korea. [10]Present address: College of Professional Services, Murdoch University, 90 South St, Murdoch, WA 6150, Australia. [11]Present address: Centre for Craniofacial & Regenerative Biology, Faculty of Dentistry, Oral and Craniofacial Sciences, King's College London, Guy's Tower, London SE1 9RT, UK. ✉e-mail: hyungchul@jnu.ac.kr; c.stern@ucl.ac.uk

non-rodent mammalian embryos can initiate primitive streak formation spontaneously at several sites, which can lead to the formation of monozygotic (identical) or conjoined twins. But this does not normally happen. What prevents twinning from occurring more frequently in these species? It has been proposed that unknown inhibitory signals emanate from the site of streak formation and that they must propagate rapidly to prevent the formation of additional streaks elsewhere, especially in large embryos such as the chick (3 mm diameter)[6–8]. When the posterior (primitive-streak-forming) side is removed from a chick embryo, the remaining anterior fragment generates a primitive streak from either its left or right posterior corner, but not both (Supplementary Fig. 1)[6,9,10]. Interestingly, GDF3/VG1 is initially expressed at both posterior corners for a short time before one of them starts to form a primitive streak (Supplementary Fig. 1), suggesting competition between the two distant extremes. However, how such competition could operate over a distance of 3 mm is unclear.

In this study, we show that calcium firing activity propagates through extraembryonic tissue via specific gap-junctions and prevents ectopic primitive streak formation and therefore twinning in chick embryos. Dynamic calcium activity and Bone Morphogenetic Protein (BMP, an inhibitor of primitive streak formation) mutually regulate each other via NF-κB and NFAT, enhancing a long-range BMP gradient. Computer simulations and experimental manipulation of calcium activity reveal that this mechanism can operate across the entire embryo and that it can establish and maintain embryo polarity to position the primitive streak and prevent formation of additional streaks. We suggest that propagation of short tracks of calcium transients regulates embryonic polarity to position the site of primitive streak formation. Such a mechanism can operate in embryos of widely different sizes. We provide evidence for similar mechanisms in two different human embryo models. Finally, we show that elements of this mechanism can also be found in *Drosophila*, suggesting that it arose early during metazoan evolution.

## Results and discussion

### Positional information travels via marginal zone gap junctions

By what route do the positional cues that prevent formation of additional primitive streaks travel across the embryo? This could occur either via the centre of the embryonic disc (area pellucida), or via the peripheral extraembryonic marginal zone (MZ), which has previously been implicated in embryonic polarity[11–13], or both (top row in Fig. 1a). To distinguish between these possibilities, we interrupted intercellular communication in the MZ by excising blocks of tissue, each spanning a little more than the width of the MZ (about 120 μm, 10–15 cells). Figure 1a summarises the experiments and predicted outcomes for excisions placed in different positions. If the positional signal travels exclusively along the MZ, excision of two blocks of MZ tissue at opposite lateral extremes of the MZ should interrupt the flow of information and mimic the removal of the posterior half of the embryo[10], leading to an additional site of primitive streak formation on either the left or the right side, just anterior to one of the two excisions. If the information travels exclusively through the MZ, ablations of left and right lateral and the extreme anterior end of the MZ should generate two additional streaks, one within each of the resulting anterior quadrants (Fig. 1a). Conversely, if the signal travels through the area pellucida, none of the three experiments should generate an ectopic streak (Fig. 1a). The results of these experiments fit the first set of predictions, consistent with the MZ being the sole route of communication of a signal for positional information (Fig. 1b and Supplementary Fig. 2a).

How does the signal travel along the MZ? We explored this by first isolating the anterior half of the embryo, followed by excision of a piece of anterior MZ at different time points (Supplementary Fig. 2b). Ectopic streak formation at the most anterior side (type3 in Supplementary Fig. 2c), implying complete repolarization, was observed more

frequently when the anterior excision was made 4–5 h or longer after the initial cut (Supplementary Fig. 2c, d). These results suggest that the positioning signals travel through the MZ across the entire embryo in less than 5 h, consistent with previous conclusions based on timed misexpression of signalling proteins[6]. The area pellucida and MZ (excluding the outer extraembryonic area opaca) in pre-primitive-streak stage embryos have an average diameter of 240 cells[14]; therefore the half-circumference of the MZ would span ~380 cell lengths, suggesting that the signal travels faster than 1 cell/min (≥1.27 cells/min). This relatively fast transmission rate suggests an intercellular mechanism based on small molecules, such as ion currents.

A previous RNA sequencing (RNAseq) analysis[14] revealed differential expression of gap-junction proteins in the MZ and area pellucida epiblast before primitive streak formation: *GJA1* (*CX43*) is enriched in the area pellucida whereas *GJB2* (*CX26*) and *GJB6* (*CX30/CX31*) are restricted to the MZ (Supplementary Fig. 3a). GJB2 and GJB6 can combine to generate functional intercellular channels, whereas GJA1 cannot participate in channel formation with GJBs (Supplementary Fig. 3b, c)[15]. This suggests that differential gap junctional communication through GJB-type channels could account for the transmission of the signal exclusively along the MZ. As a first test of this hypothesis, we cultured anterior embryo fragments (or a whole embryo with one lateral MZ excision) in 200 μM flufenamic acid, an inhibitor of all gap junctional communication[16]; this increased the incidence of formation of more than one primitive streak, suggesting that transmission of inhibitory signals had been blocked (Supplementary Fig. 3d, e). To examine whether the positional signal spreads only via GJB-type junctions, we designed a rescue experiment. Two excisions were placed at the two opposite lateral extremes of the MZ, accompanied by misexpression of either GJB2 or GJB6 in the area pellucida epiblast adjacent to one of the two excisions (Fig. 1c), to create a GJB-junction-connected route that might bypass the excision. Misexpression of either GJB2 or GJB6 rescued the marginal-zone-excision phenotype, decreasing the frequency of ectopic primitive streak formation relative to both a control plasmid and to misexpression of GJA5 (Fig. 1d, e), which should not form functional connections with GJBs (Supplementary Fig. 3b, c). Together, these results implicate gap junctional communication mediated by GJB2/6 in the MZ in the long-range transmission of positional signals that prevent ectopic primitive streak formation.

### Ca²⁺ signals between MZ cells convey positional information

Gap junctions allow intercellular passage of ions and small molecules (≤400 Da)[17]. Among the known signals that pass through this route are $Ca^{2+}$ and cyclic adenosine monophosphate (cAMP)[17,18]. First, we explored their activity by time-lapse imaging with fast fluorescent reporters. Embryos loaded with the cAMP reporter pink-Flamindo showed little activity (Supplementary Movie 1). In contrast, when embryos were loaded with the $Ca^{2+}$ indicator Cal520-AM, spontaneous $Ca^{2+}$ activity (short spikes, average 15 s full-width at half-maximum) was observed throughout the embryo but much more strongly in the MZ of pre-primitive-streak (Fig. 2ai–iii and Supplementary Movie 2) and early-streak stage embryos (Fig. 2aiv–vi and Supplementary Movie 3). Although not every cell appears to fire within a 15 min observation window, short- and medium-range wave-like propagation of $Ca^{2+}$ transients (here referred to as tracks) is seen between non-adjacent cells (Supplementary Movie 2). We explored this further using mosaic electroporation of an expression vector encoding the $Ca^{2+}$-reporter GCaMP6; high-resolution imaging revealed very thin intercellular processes connecting firing cells that were not adjacent (Fig. 2bi–iv and Supplementary Movie 4). These processes are reminiscent of the cytonemes described in *Drosophila*[19] and other systems, including the somite in early chick embryos[20]. These observations suggest that $Ca^{2+}$ spikes may be transmitted via cytoneme-like processes (Supplementary movie 4).

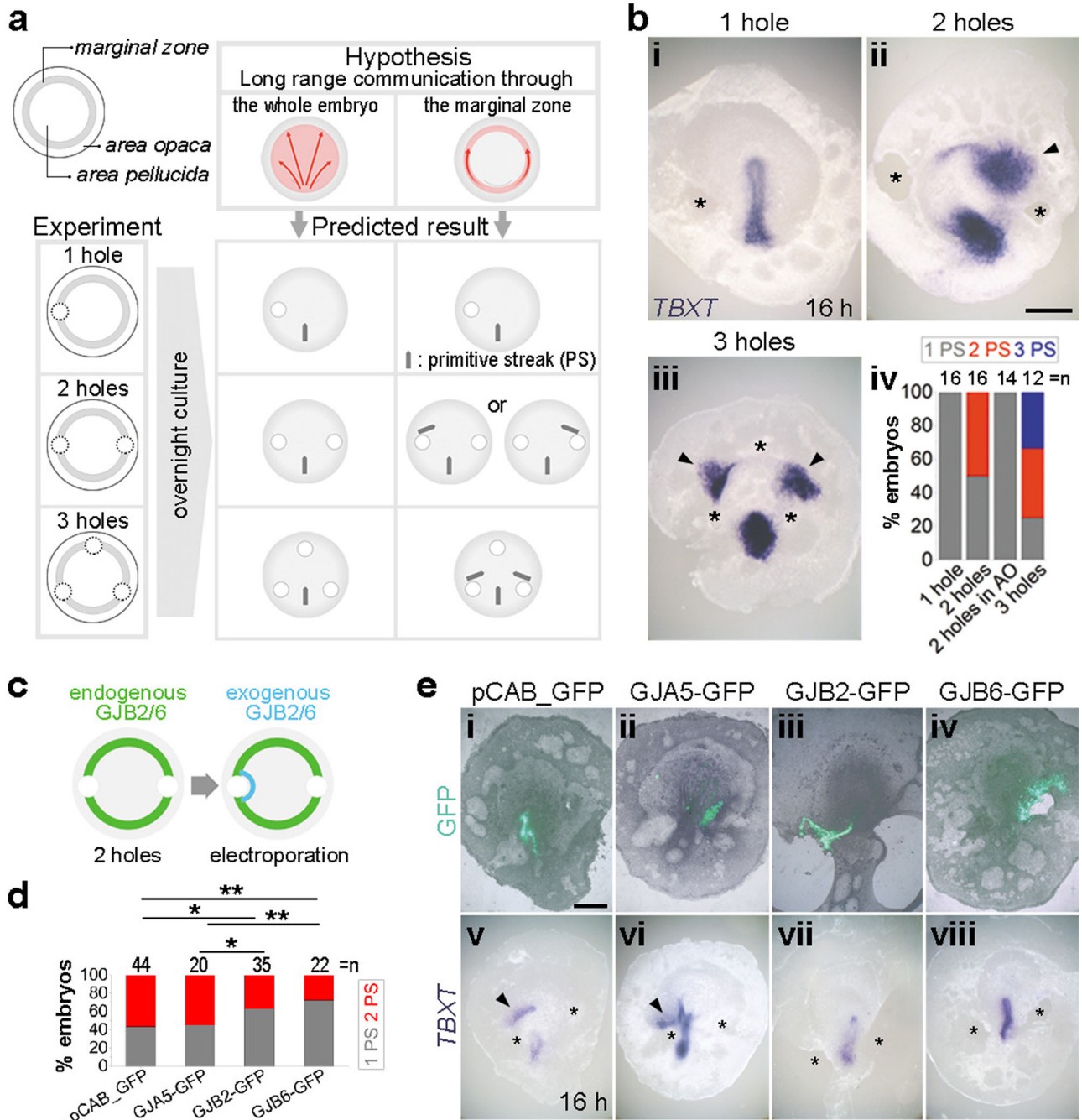

**Fig. 1 | Positional information travels along the marginal zone via specific gap junctions. a** Hypotheses and experimental design: experimental design to test two alternative hypotheses in three experiments. Top left: anatomy of the pre-primitive-streak stage embryo. **b** The marginal zone is the route of communication: formation of a primitive streak with *TBXT* (*Brachyury*) expression after overnight culture of embryos after excision (asterisks) of the marginal zone or the area opaca. **b**i–iii 1, 2 or 3 excisions in the marginal zone. Arrowheads indicate ectopic primitive streaks. Scale bar: 1 mm. **b**iv quantification of embryos with ectopic primitive streaks (control ablations in the area opaca generated no additional primitive streaks). *n* number of embryos. **c**–**e** Misexpression of GJB2/6 around the excision rescues the effect of marginal-zone excision. **c** Experimental design. Control vector

pCAB_GFP (control), GJA5-GFP, GJB2-GFP, or GJB6-GFP construct is misexpressed adjacent to the hole on one side. **d** Quantification of embryos with ectopic primitive streaks (2 PS, red). *p* values based on one-sided $\chi^2$ test with thresholds after applying a Holm−Bonferroni correction for multiple comparisons: *p* = 0.0189, 0.0051, 0.0337 and 0.0089 for bars 1–3, 1–4, 2–3 and 2–4, respectively. **e** Representative embryos with fluorescence showing misexpressed regions (**e**i–iv) and formation of primitive streak (*TBXT*) after overnight culture (**e**v–viii). The upper (GFP fluorescence) and corresponding lower (in situ hybridisation) images are of the same embryos. Asterisks: location of excision. Arrowhead: ectopic primitive streak. Scale bar: 1 mm. Source data are provided in the accompanying Source Data file.

Quantification of the distribution of firing cells in Cal520-AM-loaded embryos revealed more active cells in the MZ than in the area pellucida (AP), both anteriorly and posteriorly (Fig. 2c, d). While there is no significant difference in the frequency of firing between posterior MZ cells and adjacent posterior AP cells (average frequency:

MZ = 4.90 ± 0.24 mHz, AP = 5.26 ± 0.41 mHz), cells in the MZ fire with significantly greater amplitude (1.3-fold; *p* = 0.0189) than those in the AP (Fig. 2d). As the primitive streak appears, $Ca^{2+}$-waves increase in number and can be seen to span many adjacent cells and travel over longer distances (>200 μm), indicating that $Ca^{2+}$ activity gradually

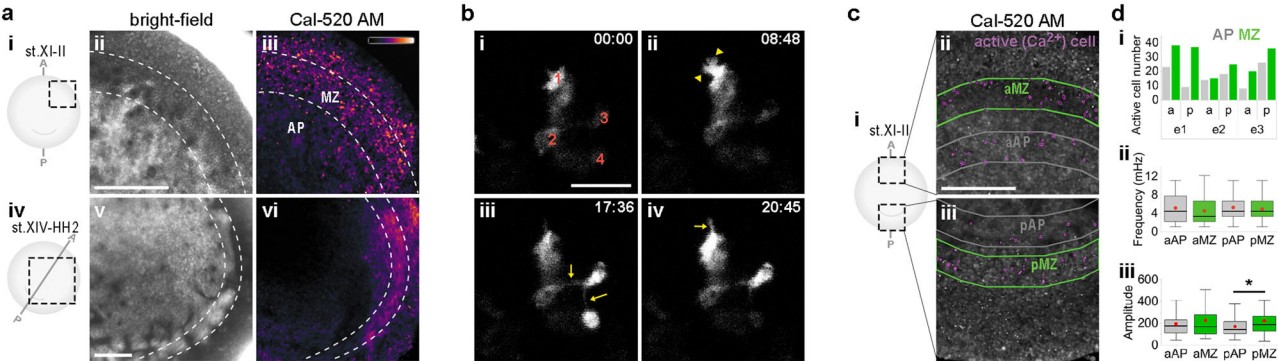

**Fig. 2 | Ca²⁺ activity in the marginal zone. a** Ca²⁺ activity in embryos at pre-primitive-streak stage (**a**i–iii) and early primitive-streak stage (**a**iv–vi). Pseudo-colour coding in (**a**iii, vi) represents relative fluorescence intensity of a maximum projection. **b** Representative time-lapse images (**b**i–iv) showing formation of cell protrusions associated with Ca²⁺ activity in cells labelled with GCaMP6. Four cells (no. 1–4, red in **b**i) are shown. Cell 1 exhibits lamellipodia (arrowheads, **b**ii) and filopodia (arrow, **b**iv). Very thin processes (cytonemes) are visible connecting cells 2–4 (arrows, **b**iii). Time is indicated in minutes. **c** Ca²⁺ activity (firing cells marked with magenta in the middle) in the marginal zone (MZ, green) and area pellucida (AP, grey). **c**i Experimental design. **c**ii, iii Representative images showing active cells. Scale bars: 500 μm in (**a**ii, **c**ii), 50 μm in (**b**ii). **d** Quantification of Ca²⁺ activity. **d**i Number of active cells (*n* = 3 embryos). **d**ii, iii Frequency (mHz) and amplitude (relative fluorescence intensity) of Ca²⁺ oscillation. *n* = 45, 73, 53 and 98 cells for aAP, aMZ, pAP and pMZ, respectively. Unpaired Student's *t* test (two-sided); *p = 0.0189. In box plots (**d**ii, iii), red dots and central lines indicate mean and median value, respectively. Box limits indicate the upper and lower quartiles, and whiskers show the range of values. Source data are provided in the Source Data file.

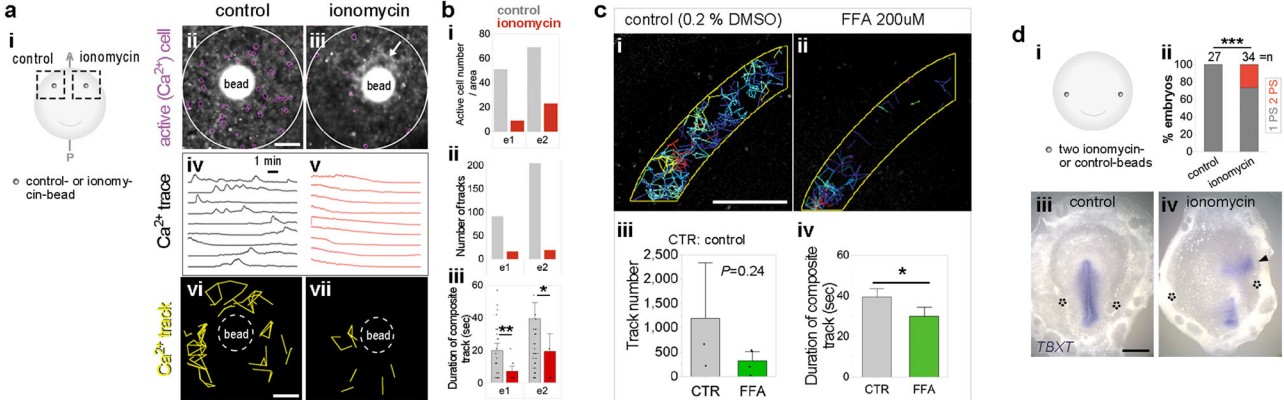

**Fig. 3 | Blockade of local Ca²⁺ activity inhibits intercellular communication. a** Ca²⁺ activity after control (0.1% DMSO) or ionomycin-bead grafts in the aMZ. **a**i Experimental design. **a**ii, iii Representative images showing the relative fluorescence intensity of a maximum projection of Cal520-AM staining. Purple circles: active (firing) cells. White arrow: increased Ca²⁺ level. **a**iv, v Representative example of Ca²⁺ traces. **a**vi, vii Tracks of intercellular Ca²⁺ activity. Mean ± s.e.m. Scale bars: 100 μm. **b**i Active-cell number. **b**ii Ca²⁺-track number (*n* = 2 embryos). **b**iii Quantification of the duration of Ca²⁺ tracks. *n* = 28 and 9 connected tracks (control and ionomycin for e1). *n* = 50 and 11 tracks (control and ionomycin for e2). Values: mean ± s.e.m. Unpaired Student's *t* test (two-sided) with Welch correction; **p = 0.0016, *p = 0.0487. **c** Ca²⁺ track analysis in the MZ of embryos after treatment of the whole embryo with flufenamic acid (FFA) compared to controls (*n* = 3 embryos). **c**i, ii Representative figures showing Ca²⁺ tracks. Scale bar: 500 μm. **c**iii, iv Quantification of the number and duration of Ca²⁺ tracks. Values: mean ± s.e.m. Unpaired Student's *t* test (two-sided); *p = 0.0207. **d** Formation of the primitive streak after two control or ionomycin-bead grafts in the lateral MZ. **d**i Experimental design. **d**ii Summary of results. Boschloo's test (one-sided); ***p = 0.0007. **d**iii, iv Representative phenotypes. Dotted circles: location of the beads. Arrowhead: ectopic primitive streak. TBXT: primitive streak marker. Scale bar: 1 mm. Source data are provided in the Source Data file.

increases during primitive streak formation (Supplementary Movie 3 and Supplementary Fig. 4).

If an intercellular Ca²⁺ signal is required for the gap-junction-mediated transmission of positional information in the MZ, local blocking of Ca²⁺ activity should mimic the effect of MZ-excision (see Fig. 1a, b). The Ca²⁺-ionophore ionomycin blocks intercellular communication by flooding cells with Ca²⁺[21]. An ionomycin-bead graft in the anterior-lateral MZ increased the basal level of Ca²⁺ in nearby cells (Fig. 3aiii, arrow), but strongly repressed local Ca²⁺-activity (Fig. 3a, b), including the number of firing cells (by 3–6-fold), and of intercellular Ca²⁺-tracks (6–11-fold) as well as the duration of propagation of the tracks (2–3-fold) near the bead (Fig. 3a, b), compared to control (see Methods and Supplementary Fig. 5). A similar decrease in Ca²⁺-tracks (Fig. 3ci–iv) was seen after treatment with the gap junction blocker

flufenamic acid (FFA), confirming that the Ca²⁺ activity propagates via gap junctions. To examine the effect of blocking intercellular Ca²⁺ transport on embryonic polarity, two ionomycin-beads were grafted into the left and the right lateral MZ, respectively (Fig. 3d). The consequences resembled the marginal-zone-excision experiment (Fig. 1), with a single ectopic primitive streak forming near one of the beads (Fig. 3h and Supplementary Table 1).

## Local Ca²⁺ signals set up a long-range positional gradient

The above results suggest that relatively short-range transmission of Ca²⁺-transients (tracks) conveys positional information over a large embryonic territory, to regulate the site of primitive streak formation. How could local propagation of Ca²⁺-signals provide a long-range positional cue? Since Ca²⁺ waves were not seen to travel long distances

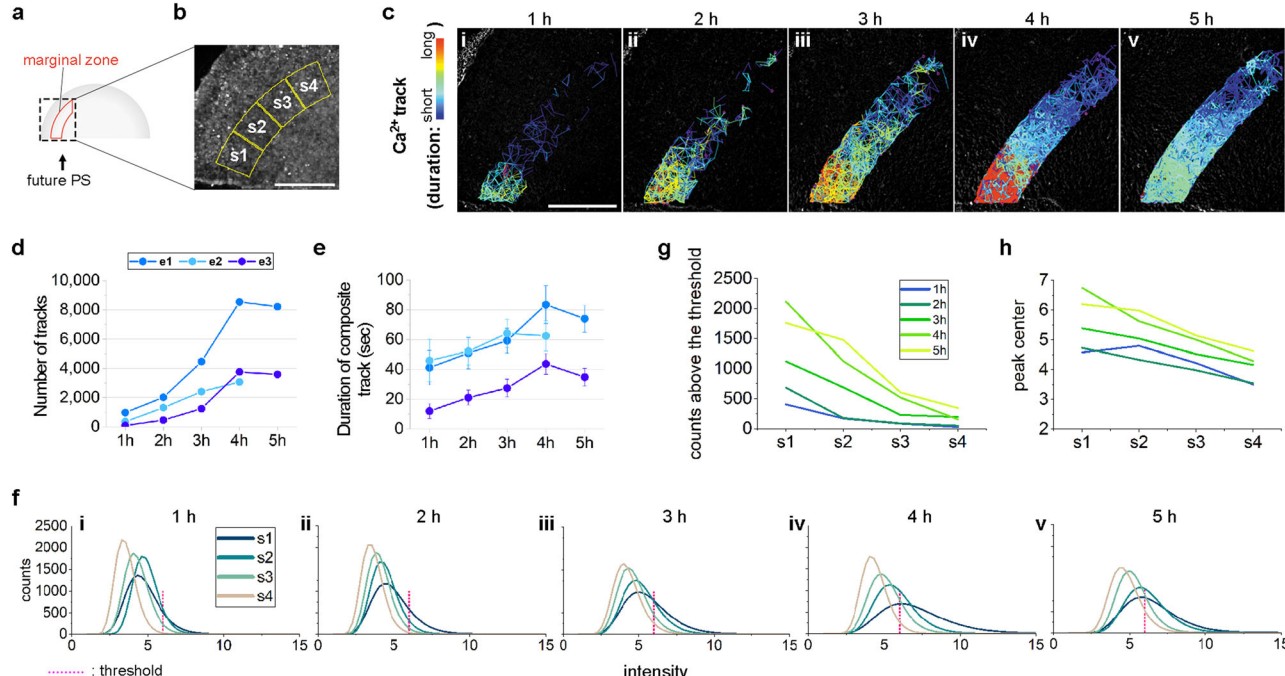

**Fig. 4 | Gradual propagation of Ca²⁺ activity during initiation of the primitive-streak-forming process. a, b** Experimental design. The primitive-streak (PS)-forming side of an isolated anterior half was monitored every 1 h for Ca²⁺ activity—the marginal zone (red) was analysed for Ca²⁺ tracks (**a**, **c**–**e**) or regional differences in Ca²⁺ activity (**b**, **f**–**h**). *TBXT* expression was checked after overnight culture to determine from which side the PS formed. Dashed square: region analysed (shown in detail in **b**, Scale bar: 500 μm). **c** Representative images (e1 out of three embryos named e1–e3) showing the increase in tracks of intercellular Ca²⁺ firing activity in the marginal zone of the PS-forming side over time (1 h intervals) (**ci**–v). Pseudo-colour coding represents the duration of the tracks. Scale bar: 500 μm.

Quantification of number (**d**) and duration (**e**) of Ca²⁺ tracks. Note that e2 was monitored only for 4 h. Details of the number of tracks are in 'Methods'. Values: mean ± s.e.m. **f**–**h** Histogram analysis (e1) showing a spatial gradient of Ca²⁺ activity with temporal increase (see Supplementary Fig. 7 for e2 and e3). **f** Histogram and comparison of four regions (s1–4 as in **b**) at every 1 h (**fi**–v). *x*-axis: relative fluorescence intensity. *y*-axis: pixel counts. **g** Difference in pixel counts above the threshold (magenta dotted line in **f**). **h** Changes in peak centre position (highest point of each graph in **f**), revealing a temporal increase in the gradient of Ca²⁺ activity. The threshold is set at 6, corresponding to ~10% of the total counts of e1 at 1 h. Source data are provided in the Source Data file.

before primitive streak formation (Supplementary Movie 2), we hypothesised that a gradient of Ca²⁺ activity, rather than Ca²⁺ concentration, along the MZ might act as the positional cue. To observe the propagation of firing activity, one edge (left or right) of an isolated anterior-half blastoderm obtained by cutting a Cal520-AM-loaded embryo was imaged for 10 min every 1 h for several hours. The position of the primitive streak was recorded after overnight culture. Analysis of the movies (Fig. 4a, c) revealed gradual propagation of short-range Ca²⁺ tracks along the MZ, arising from the primitive-streak-forming side (Fig. 4c); no such propagation was observed in the area opaca or area pellucida (Supplementary Fig. 6a, b). A gradual increase in both the number and duration of Ca²⁺ tracks was observed in the first 4 h, before decreasing slightly again (Fig. 4d, e). The direction of each track seemed to be random rather than biased (Supplementary Fig. 6h, i). The non-primitive-streak-forming side displayed a different pattern of Ca²⁺ tracks: here they first decreased in number and duration before increasing again (Supplementary Fig. 6c–e). To check whether there is a gradient of Ca²⁺ activity along the MZ, the MZ was divided into four regions (S1–S4) and their Ca²⁺-activity was compared (Fig. 4b). This revealed both temporal and spatial gradients of Ca²⁺ activity along the MZ (Fig. 4f and Supplementary Fig. 7a). The highest Ca²⁺ activity was observed very close to the cut edge at the future primitive-streak-forming side (Fig. 4g, h and Supplementary Fig. 7c, d), whereas the non-primitive-streak-forming side of the cut edge exhibited rapid dampening and a new increase in Ca²⁺ activity without a spatial gradient (Supplementary Fig. 7b, e, f). Together, our data suggest that, as the primitive streak starts to form, a wave of increasing Ca²⁺-activity propagates through gap junctions along the MZ, generating a spatial gradient of Ca²⁺-activity.

## Cross-talk between BMP and Ca²⁺ regulates embryo polarity

As in amphibian and fish embryos, the site of gastrulation in amniote embryos is determined by the balance between inducing (GDF1/GDF3/Nodal-like, Smad2/3-activating) and inhibitory (BMP-like, Smad1/5/8-activating) signals at opposite extremes of the embryo[6,7,22–26]. Application of either cells expressing Chordin[27,28] or of a bead of dorsomorphin (DM, a BMP inhibitor)[29] to the chick anterior MZ at pre-primitive-streak stages can induce an ectopic primitive streak next to the bead (Supplementary Fig. 8a, b). The bead causes a local decrease in nuclear localisation of pSMAD1/5/8 (Supplementary Fig. 8h), confirming that it does inhibit BMP. DM treatment of a whole embryo causes multiple primitive streaks (expressing *TBXT*) to form, which is preceded by ring-like expression of *GDF3/VG1* all around the MZ, as well as reduced expression of *GATA2* (a downstream target of BMP signalling) (Supplementary Fig. 8c). These results confirm a conserved role of BMP in embryonic polarity in the chick embryo, inhibiting primitive streak formation[5,7,27]. Next, we investigated whether inhibiting BMP with DM can regulate Ca²⁺ activity. Grafting a DM-bead only slightly increases the incidence of Ca²⁺-active cells near the bead in a subset of embryos; however, stronger, statistically significant effects are seen in the frequency (which is reduced; $p = 5.928 \times 10^{-11}$) and the amplitude (increased; $p = 7.686 \times 10^{-5}$) of Ca²⁺-oscillations relative to controls grafted with a DMSO-bead (Fig. 5a, b). Moreover, these effects of DM bead implantation are comparable with the properties of Ca²⁺-firing at the posterior side of normal embryos (see Fig. 2c, d). The DM-bead graft also increased the number and duration of Ca²⁺ tracks near the bead (Supplementary Fig. 6f). Do primitive-streak-inducing signals (GDF3/Vg1/Nodal/Activin) have the same effect? To test this, we implanted an ACTIVIN-bead[29,30] to the anterior MZ. Like with DM

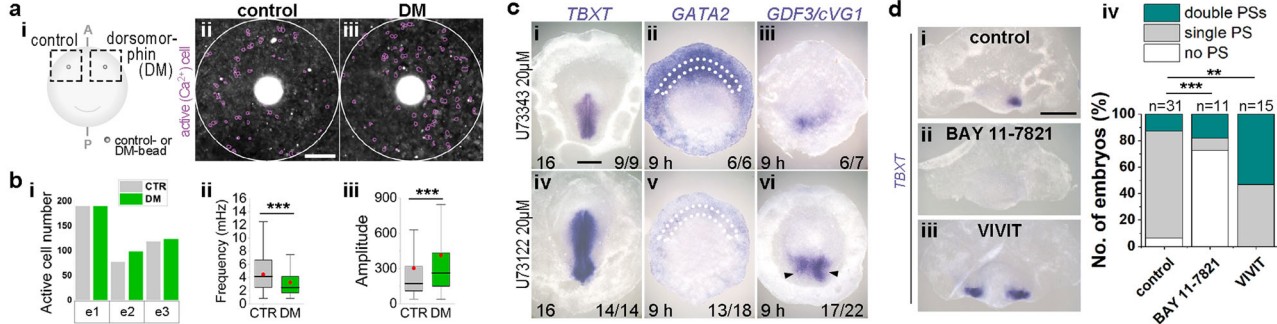

**Fig. 5 | Ca²⁺-based mechanism for positioning the primitive streak. a–c** Cross-regulation between BMP and Ca²⁺. **a, b** Ca²⁺ activity after control- or dorsomorphin (DM)-bead grafts in the aMZ. **ai** Experimental design. **aii, iii** Representative images showing relative fluorescence intensity of Cal520-AM staining (maximum projection). Purple circles: active (firing) cells. Scale bar: 100 μm **bi** Active cell number (*n* = 3 embryos). Two of the embryos (e2 and e3) show more active cells in DM-treated regions relative to control regions. **bii, iii** Frequency and amplitude (relative fluorescent intensity) of the Ca²⁺ oscillations. *n* = 386 and 411 cells for control and DM, respectively. Unpaired Student's *t* test (two-sided); ***$p = 5.9 \times 10^{-11}$ and $7.7 \times 10^{-5}$, respectively. In box plots, red dots and central lines indicate mean and median. Box limits indicate upper and lower quartiles; whiskers show range. **c** Effect of treatment of whole embryos with a control (U73343, structural analogue of

U73122) (**ci–iii**) or U73122 (**civ–vi**) on expression of *TBXT* (PS-formation), *GATA2* (a target of BMP signalling) and *GDF3*. Incubation time is indicated. Fractions represent number of embryos with similar results to the phenotype shown: for *TBXT* and *GDF3/cVG1*, normal (upper) or enlarged/multiple (lower) expression; for *GATA2*, anterior (upper) or no (lower) expression. White dotted lines: aMZ. Arrowheads: multiple PS. Scale bar: 1 mm. **d** Opposite roles of NF-κB and NFAT on primitive streak formation. Effect of control (0.05% DMSO), BAY 11-7821 (NF-κB inhibitor), or VIVIT (NFAT inhibitor) treatment of isolated anterior half-embryos on PS formation (*TBXT*). **di–iii** Representative images for each condition. Scale bar: 1 mm. **div** A summary graph of the number of embryos with different morphology. BAY 11-7821 inhibits streak formation, while VIVIT promotes streak formation. Boschloo's test (one-sided); ***$p = 1.7 \times 10^{-5}$ and **$p = 0.0031$.

beads, this increased the number of Ca²⁺-active cells near the bead, but unlike DM-beads, the amplitude of the Ca²⁺-oscillations was reduced, and there was no change in frequency compared to controls (Supplementary Fig. 6g). Therefore, Ca²⁺-activity seems to be modulated primarily by BMP during primitive streak formation.

How, then, does the long-range travelling Ca²⁺ activity influence primitive streak formation? One possibility is that Ca²⁺ activity and BMP cross-regulate each other. To test this, whole embryos were grown in the presence of 20 μM U73122, a phospholipase-C inhibitor that blocks inositol 1,4,5-trisphosphate (IP₃)-mediated Ca²⁺ mobilisation from intracellular Ca²⁺ stores[31]. U73122 treatment caused much larger or multiple primitive-streaks to form, while its inactive structural analogue U73343[31] had no effect (Fig. 5ci, iv). Similarly, in dorsal-ventral patterning of zebrafish embryos, it was previously reported that treatment with 2-aminoethoxydiphenyl borate (2-APB), an IP₃ receptor antagonist, causes massive expansion of dorsal tissue, shown by enlargement of the domain expressing *goosecoid*[32]. In our experiments, U73122 also downregulated *GATA2* in the MZ, along with upregulation of *GDF3/VG1* expression at multiple sites around the circumference (Fig. 5cii, iii, v, vi). These results suggest that Ca²⁺ activity acts as an inhibitory signal by regulating BMP activity and thereby primitive-streak formation, probably involving IP₃ and intracellular Ca²⁺ stores. In contrast, ionomycin treatment of whole embryos has the opposite effect, inhibiting primitive streak formation and *GDF3/VG1* expression, while upregulating *GATA2* expression (Supplementary Fig. 8d, e). Conversely to U73122, treatment of whole embryos with modulators of cellular Ca²⁺ that act in other ways, such as nicardipine (Ca²⁺-channel blocker) or FFA (gap-junction blocker), inhibited primitive-streak formation (Supplementary Fig. 8f) without affecting embryo expansion or growth. These results further support the idea that Ca²⁺ regulates BMP activity via IP₃ and intracellular Ca²⁺ stores.

### Cross-regulation between Ca²⁺ and BMP via NF-κB and NFAT
By what mechanism does the Ca²⁺ signal regulate BMP activity to generate and/or maintain a BMP gradient along the marginal zone? A previous study showed that different properties of Ca²⁺ signalling can activate different transcription factors: large Ca²⁺ transients selectively activate nuclear factor kappa-light-chain-enhancer of activated B-cells (NF-κB), while a low sustained plateau of Ca²⁺ activates nuclear factor of activated T-cells (NFAT)[33]. Moreover, NF-κB inhibits BMP activity[34,35],

while NFAT can act as a positive regulator of BMP[36,37]. We therefore hypothesise that different Ca²⁺ activities regulate BMP by activating either NF-κB or NFAT. To test whether NF-κB and NFAT are involved in positioning the primitive streak in the early chick embryo, we first treated anterior half-embryos either with the NF-κB inhibitor BAY11-7821 or with the NFAT inhibitor VIVIT (Fig. 5d). Inhibition of NF-κB blocked primitive streak formation, while the NFAT inhibitor induced a primitive streak from both left and right edges at significantly higher frequency than controls (Fig. 5d). These results support the hypothesis that two opposing signals, high calcium activity/NF-κB/inhibition of BMP at one end, antagonised by low calcium activity/NFAT/activation of BMP at the other, respectively promote and inhibit primitive streak formation. Indeed, inhibition of BMP by grafting a DM-bead caused a large increase in NF-κB expression near the DM bead as well as a reduction in the level of NFAT expression near the bead (Supplementary Fig. 8h). These results suggest that NF-κB and NFAT are involved in the cross-regulation of BMP and Ca²⁺. We therefore propose that a gradient of Ca²⁺-activity (rather than absolute levels of Ca²⁺) in the MZ regulates BMP signalling through NF-κB and NFAT to position the primitive streak: in the posterior MZ, high Ca²⁺ activity upregulates NF-κB, enabling primitive streak formation by inhibiting BMP; anteriorly, low Ca²⁺ activity induces NFAT, increasing BMP activity and blocking primitive streak formation. This is consistent with our finding that sustained elevation of Ca²⁺ by an ionomycin bead induces expression of the BMP target *GATA2* near the bead (see above and Supplementary Fig. 8e).

This mechanism involving calcium, NF-κB and NFAT regulating BMP levels operates cell-autonomously. However, only a subset of MZ cells show calcium activity. It has been shown in many systems that BMP activity positively regulates BMP production (in a paracrine way)[38–41]. We therefore propose that calcium firing regulates BMP production and/or its degradation cell autonomously in some cells, which then travels to neighbouring cells to smoothen the local BMP concentration irrespective of whether all of those cells are firing calcium or not. Effectively, this mechanism translates short-range tracks of Ca²⁺-firing cells into a wave of calcium activity that travels along the MZ of the embryo, generating and/or maintaining a stable BMP concentration gradient.

### Computer model for long-range competition
To explore the dynamics of how the two extremes of either normal embryos or of an isolated anterior half-embryo communicate with

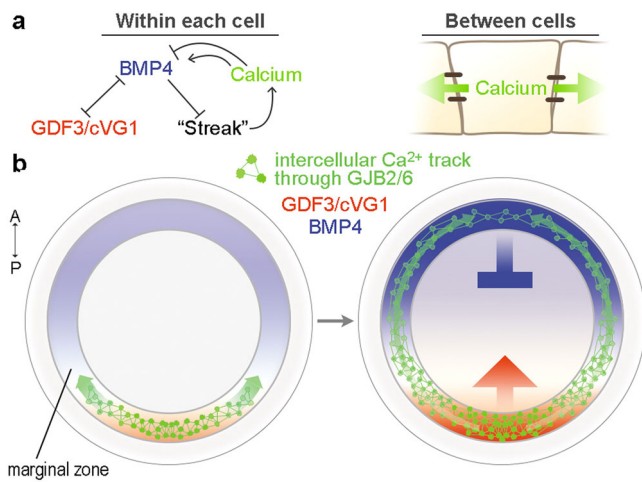

**Fig. 6 | Proposed mechanism for positioning the primitive streak. a** Interactions included in the computational model to simulate primitive streak positioning. Left: interactions between the components in the model. Right: the calcium activity conveying the streak-inhibiting signal travels across the marginal zone cells. **b** Schematic diagram of the proposed mechanism of embryonic polarisation and primitive streak formation. Initially, GDF3 and $Ca^{2+}$ activity increase where BMP4 is low. $Ca^{2+}$ travels along an extraembryonic region (the marginal zone) via specific gap junctional communication, making its spatial gradient. Lower $Ca^{2+}$ activity in the anterior region upregulates inhibitory signals on primitive streak formation (BMP), while higher $Ca^{2+}$ activity in the posterior region inhibits BMP activity. Source data are provided in the Source Data file.

each other to establish a stable axis of polarity, we used computer simulations. The MZ is modelled as a one-cell-thick ring of cells (x-axis of the graph in Supplementary Fig. 9). For each cell, we define levels of BMP4, GDF3/VG1 and '$Ca^{2+}$-activity', which activate/inhibit cells adopting a 'streak identity' (left, Fig. 6a). The $Ca^{2+}$ activity can transmit quickly between cells (right, Fig. 6a). The system is governed by the level of BMP4, with an initial gradient decreasing posteriorly, mimicking that found in the pre-streak embryo. In the absence of $Ca^{2+}$activity, BMP4 decays linearly: low $Ca^{2+}$ activity is required for BMP4 production (Supplementary Fig. 9 and 'Methods'). As BMP4 drops below a first threshold, GDF3/VG1 is induced. As BMP4 continues to decay and passes a second threshold, 'streak identity' is induced (Supplementary Fig. 9; $t = -0.06$ h). Streak cells then produce $Ca^{2+}$ spikes that transmit quickly between cells, establishing a gradient of calcium activity, decreasing anteriorly. The gradient of $Ca^{2+}$ activity maintains the spatial gradient of BMP4 by degrading BMP4 posteriorly, where $Ca^{2+}$ activity is high, while activating BMP4 anteriorly where $Ca^{2+}$ activity is low. These conditions result in an embryo with a single primitive streak situated posteriorly, a gradient of $Ca^{2+}$ activity highest posteriorly and an opposite gradient of BMP4, as observed in intact embryos (Supplementary Fig. 9; $t < 0$, Supplementary Movie 5).

Next, we applied the model to the situation of an isolated anterior half-embryo to explore how such fragments become stably polarised (Supplementary Fig. 9; $t > 0$, Supplementary Movie 6). We model a slightly oblique cut, with the right side leaning posteriorly, to generate an initial bias. Removal of cells with streak identity causes $Ca^{2+}$ activity to decay quickly, followed by a fall in the level of BMP4, since low $Ca^{2+}$ activity is required to maintain BMP4. This results in asymmetric induction of GDF3/VG1 at both left and right posterior edges of the anterior half-embryo, followed by a rapid increase in $Ca^{2+}$ activity on the right side (Supplementary Fig. 9; $t = 3.04$ h). $Ca^{2+}$ activity is sensitive to local BMP4 levels and propagates across the half-embryo, establishing gradients and competition between the two extremes. As a result, BMP4 levels fall on the right side, causing 'streak identity' to be established on that side. This model mimics two key behaviours

observed in an isolated anterior half: transient expression of *GDF3/VG1* on both sides, resolving to the formation of a single primitive streak and reorientation of the BMP4 gradient (Supplementary Fig. 1a).

The above results suggest a mechanism for how embryos position the primitive streak via long-range communication involving $Ca^{2+}$ (Fig. 6b). At the start of the streak-forming process, $Ca^{2+}$ activity increases in a subset of cells of the MZ in areas with low BMP activity and propagates along the MZ. At the same time, $Ca^{2+}$ activity feeds back within the cell to regulate BMP activity, which in turn is transmitted to its neighbouring cells. This generates a spatial gradient of $Ca^{2+}$ activity in the MZ, which helps to maintain a more uniform gradient of BMP activity. Thus, the overall speed of communication in the embryo is regulated by combinatorial action between $Ca^{2+}$ and BMP.

### Conservation in human and non-human animals

Is this $Ca^{2+}$-based positioning system conserved in other species? In zebrafish (non-amniote vertebrate) embryos, calcium activity has been observed before gastrulation, biased to the dorsal region, where the embryonic shield (the fish equivalent to the primitive streak of amniotes) will arise[32,42]. Consistent with our findings in the chick, treatment with U73122 or 2-APB decreases the number of $Ca^{2+}$ transients in the dorsal region; the latter also causes expansion of *goosecoid* (a marker of the shield)[32]. To investigate whether a $Ca^{2+}$ signalling system might be present in early human embryos, we also examined the development of stem cell-derived human embryoids that can develop polarity; we used two different human embryo models. First, we investigated whether $Ca^{2+}$-activity is involved in polarisation of human embryoids. A microfluidic amniotic sac embryoid assay (post-implantation amniotic sac embryoids, PASE) allows BMP4 treatment to be applied to one side of the spherical embryoid, which causes primitive-streak-like cells expressing BRA (Brachyury/T) to form on the opposite side, whereas no polarisation takes place in the absence of BMP4 (Supplementary Fig. 10a)[43]. To investigate whether $Ca^{2+}$ activity accompanies the polarisation during PASE development, Cal520-AM was loaded into this system. Only BMP4-treated PASEs showed active $Ca^{2+}$ activity in primitive-streak-like cells in the pole opposite to BMP4 treatment, and no $Ca^{2+}$ activity was observed in an embryoid without BMP4 treatment (Supplementary Movie 7). This result suggests that, as in the chick, BMP4 regulates $Ca^{2+}$ activity during the polarisation of the PASE.

To determine whether calcium regulates the polarisation of human embryoids, we used another stem cell-based human embryo model system (sometimes referred to as gastruloids) in which 3-dimensional aggregates of human embryonic stem cells surrounded by medium spontaneously polarise and become elongated, expressing markers for mesoderm (BRA), neuroectoderm (SOX2) and endoderm (SOX17) at one end, marking the so-called posterior (primitive-streak) side of the aggregate[44]. Inclusion of 20 μM U73122 in the culture medium rapidly increased the expression of all three markers (1.5 h) in the entire embryoid relative to DMSO- or U73343-treated controls (Supplementary Fig. 10b). The difference became more marked with longer incubation (3.5 h) (Fig. 7a, b), comparable to the results with chick embryos (Supplementary Fig. 8c). Together, our results in chick embryos and human embryoids, along with previous findings in zebrafish, suggest that reciprocal regulation of BMP and $Ca^{2+}$ may represent a conserved mechanism for positioning the site of gastrulation in vertebrates, from fish to human.

The relationship between BMP/TGFβ and NF-κB seems to have ancient evolutionary origins: in the fruit fly *Drosophila*, opposing gradients of *dorsal* (the fly ortholog of NF-κB) and *decapentaplegic* (*dpp*) (the main TGFβ/BMP homologue in fly) establish the dorsoventral axis and thereby position the site of gastrulation (ventrally)[45–47]. Although only a few calcium-firing cells are scattered around the embryo, their number increases at around the time of onset of gastrulation, suggesting a possible role of $Ca^{2+}$ signalling in initiating gastrulation[48]. To

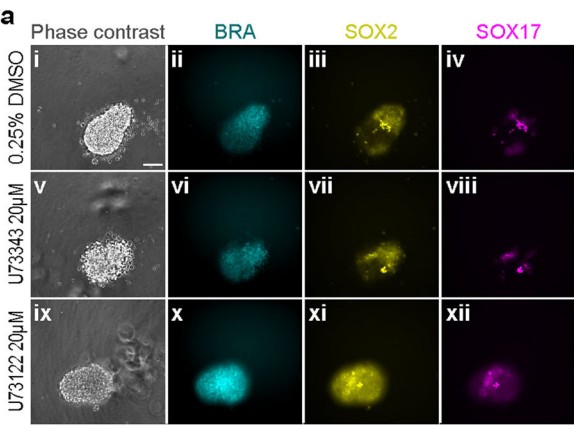

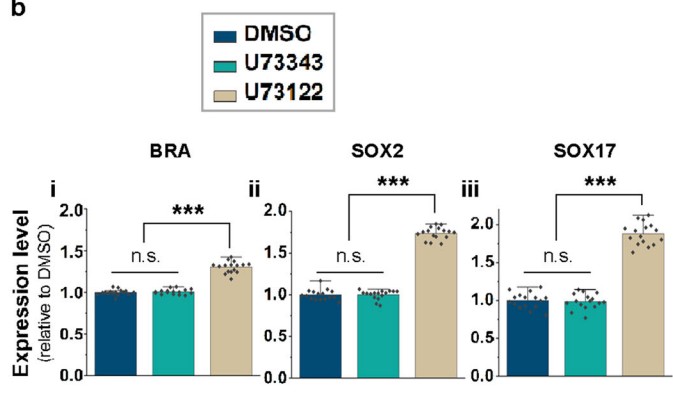

**Fig. 7 | Effect of inhibiting PLC on polarisation of human embryoids. a** Effect of DMSO (**ai–iv**), U73343 (**av–viii**) or U73122 (**aix–xii**) to human embryoids on the expression of markers of mesoderm (BRA, cyan), neuroectoderm (SOX2, yellow) and endoderm (SOX17, magenta) using fluorescent reporters. **a** Representative images of the embryoids, 3 h after chemical treatment. Scale bar: 100 μm.

**b** Expression levels of BRA (**bi**), SOX2 (**bii**) and SOX17 (**biii**) 3 h after chemical treatment (n = 16 for each). Values: mean ± s.e.m. One-way ANOVA; ***p = $1.07 \times 10^{-23}$, $1.77 \times 10^{-34}$, and $1.15 \times 10^{-26}$, respectively. Bonferroni correction for multiple comparisons. n.s. not significant. Source data are provided in the Source Data file.

study the role of Ca²⁺ signalling in *Drosophila* gastrulation, we treated early *Drosophila* embryos with U73122: this caused a significant (p = 0.0462; Fisher's exact test, n = 56) delay in the initiation of gastrulation (without affecting the progression of cellularisation; p = 0.0754; Fisher's exact test) compared to control embryos treated with U73343 (n = 34), suggesting that Ca²⁺ activity may also be involved in regulating the initiation of gastrulation in the fly (Supplementary Fig. 10c). Together, these results strongly implicate calcium signalling in the regulation of embryonic polarity by positioning the site where gastrulation will begin, which appears to have ancient evolutionary origins and seems to be conserved in humans. Moreover, this mechanism provides a solution to the long-standing problem of how long-range gradients of morphogens can be established across large developing systems, and also explains how they can be scaled appropriately to pattern embryos of different sizes. Recently, a similar mechanism was proposed in which a long-range gradient of a morphogen (e.g. Wnt) is established by cell-to-cell signalling relay (like that proposed for Ca²⁺ in our study) during planarian regeneration[49]. This suggests that short-range propagation of activities like Ca²⁺ firing might be a general mechanism for the establishment and maintenance of long-range gradients of positional information without relying on simple diffusion of a morphogen over long distances.

## Methods

### Chick embryo culture, manipulation, and wholemount in situ hybridisation

Fertilised White Leghorn (*Gallus gallus domesticus*) hens' eggs (Henry Stewart, UK) were pre-incubated for 2 h, then the embryos were harvested in Pannett-Compton saline (PCS)[50] and cultured for the desired time by a modified New culture method[51,52]. Unhealthy and abnormal embryos were discarded before setting up the culture. To make a hole in the embryo, the excision was made using a bent insect pin before culture. Whole mount in situ hybridisation was conducted as previously described[53,54]. The probes used were: *BMP4*[55], *GATA2*[56], *GDF3* (*cVG1*)[24], *GJA1* (S. Price), *GJB1* (chEST13M9), *GJB2* (full-length coding sequence in pGEM-T-easy), *GJB6* (chEST89h10), *NODAL*[57], *PITX2*[58], *TBX6*[10], *TBXT (Brachyury)*[59]. Stained embryos were imaged with an Olympus SZH10 stereomicroscope with a QImaging Retiga 2000R camera.

### Proteins and chemicals

For local treatment with chemicals or proteins, AG1X2-formate beads (for chemicals) or Affigel Blue beads (BIO-RAD, 1537302; for BMP4)

were soaked in different concentrations of the desired protein or chemical overnight at 4 °C. Beads were washed in PCS before grafting. Dimethyl sulfoxide (DMSO, 0.2%) or BSA (0.1%) was used to dilute the chemicals or proteins, respectively, and for soaking the control beads. Final concentrations used for microbead-soaking: 50 ng/μl recombinant human BMP4 (R&D systems, 312-BP), 200 μM dorsomorphin dihydrochloride (Tocris, 3093), 2 μM ionomycin (Sigma, I9657). For chemical treatment to the whole embryo, the chemical was diluted first in PBS (1:10 v:v) and then in egg albumen (9:10 v:v), which was used to culture the embryos (under the vitelline membrane). For treatments with VIVIT (Tocris, 3930) and BAY 11-7821 (Tocris, 1744), embryos were first soaked in the chemical diluted in PCS for 1 h, prior to culture with albumen containing the same concentration of the chemical. Final concentrations used for treatment of whole embryos: 20 μM dorsomorphin, 200 μM flufenamic acid (Sigma, F9005), 2 μM ionomycin, 50 μM, nicardipine (Sigma, N7510), 20 μM U73122 (Sigma, U6756), 20 μM U73343 (Sigma, U6881), 12 μM VIVIT (Tocris, 3930), 12.5 μM BAY 11-7821 (Tocris, 1744).

### Chicken connexin modelling

The protein sequences of chicken connexins were BLASTed[60] against the RCSB PDB protein data bank (https://rcsb.org)[61] to find close homologues with known structures. Modeller 9.24 (https://salilab.org/modeller/)[62] was used to align chicken Cx sequences to the templates from the BLAST search (pdb code 2zw3 for GJB2 and GJB6; 2zw3, 5er7 and 1r5s for GJA1; 2k7m, 2zw3 and 6mhy for GJA5), which generated 100 structures per chicken connexin. The ones with the best DOPE score were selected for this study. PyMol v1.74 was used for graphical representations and for structural analysis.

### Misexpression of connexins

For misexpression of different connexin proteins, expression vectors conjugated with green fluorescent protein were used. Control, pCAβ-IRES-GFP. GJB6, Cx30-msfGFP (Addgene plasmid #69019)[63]. For GJB2, chicken GJB2 (NM_001270816.1) was amplified from cDNA and inserted into pCAβ-IRES-GFP. PCR amplification was carried out using PCR Biosystems Ultramix (PCR Biosystems). Briefly, a 25 μl reaction mixture contained 1 μl cDNA template (from the appropriate HH stage, <100 ng), 1–2.5 μl Forward and Reverse primers (10 μM), 12.5 μl PCR Bio 2X Ultramix and PCR grade H₂O. Cycle conditions followed the manufacturer's guidelines. All PCR amplified fragments were sub-cloned into pGEM-T Easy (Promega, A1360) and sequences were confirmed prior to sub-cloning. PCR primers used were: BamHI_forward_Cx26,

5′-GGAAATgGATCCTTTGCTGCTTGG-3′; NotI_reverse_Cx26, 5′-GGGAA gcggccgcTTACTTTAA-3′. After excision, misexpression of the expression vectors near the hole was conducted by electroporation with 3 μg/μl of DNA as previously described[64].

### Live imaging and Ca²⁺ signal analysis of chick embryos

Embryos were harvested in PCS, then incubated in 10 μM Cal-520 AM (Abcam, ab171868) in PCS containing 0.02 % Pluronic F-127 (Sigma, P2443) for 2 h at 37 °C. They were washed in PCS and cultured for 30 min to 1 h to settle down on the membrane. To monitor cellular protrusions associated with Ca²⁺ activity, another Ca²⁺ indicator, GCaMP6s[65] was electroporated in the anterior marginal zone. Ca²⁺ activity was imaged in live embryos with an inverted Leica TCS SPE confocal microscope. To check cAMP activity in live embryos, anterior marginal zone of the embryo was electroporated with the cAMP indicator Pink Flamindo[66] and imaged with a Zeiss 880 Airyscan microscope using excitation = 567 nm and emission = 590 nm.

After recording of the movies, the CaImAn software[67,68] was used to count the number of firing cells ('Active cells' in Fig. 2) and to analyse the Ca²⁺ traces for motion correction, segmentation and source extraction. Motion correction was performed using the NoRMCorre algorithm. Constrained nonnegative matrix factorization (CNMF) was performed using the 'greedy_roi' initialisation, patches of 48 × 48 pixels and a gSig value of 6 based on estimated cell diameters. The resulting Ca²⁺ traces from CaImAn were then analysed to measure the frequency and amplitude of Ca²⁺ transients using the Peak Analyzer wizard of OriginPro software (2019, OriginLab Corporation). Baseline smoothing was done by the asymmetric least squares method: asymmetric factor, 0.001; threshold, 0.02; smoothing factor, 4; number of iterations, 10; auto subtract-baseline and rescale. Peak finding settings: smoothing window size, 20; local maximum with 5 local points; filtering, threshold height = 10%.

### Analysis of Ca²⁺ tracks, angle distribution, and activity

In these experiments, three embryos were imaged at the future PS forming side (e1–e3) and two embryos (e4–e5) were imaged at the non-PS forming side. Analysis of tracks of Ca²⁺ firing activity was conducted separately for each selected region (area opaca, marginal zone or area pellucida) and was performed at the single-cell level (which involved identifying a fluorescent cell in one frame, then another firing cell within a certain distance to the first cell in the next frame, and linking them), using the TrackMate plugin[69] in Fiji[70]. Before TrackMate analysis, 200 frames (10 min) of the Ca²⁺ movie were analysed in Fiji to get 100 x ΔF/F0 values (F0 = average Z projection). Cells were detected by the LoG detector (threshold, 13; estimated blob diameter, 13 μm) with median filtering and subpixel localisation. Then, tracks were extrapolated using LAP tracker (max distance: frame linking, 90 μm; gap closing, 90 μm within 2 frames; splitting, 90 μm) (Supplementary Fig. 5). To estimate the duration of the Ca²⁺ tracks, the following numbers of connected tracks were analysed at each time point: for the PS forming side, e1, n = 95, 136, 244, 294, 322; e2, n = 64, 125, 163, 188; e3, n = 51, 100, 169, 269, 317, and for the non-PS forming side, e4, n = 197, 53, 55, 64, 178 and e5, n = 57, 30, 133, 295, 334. To investigate angle distribution, the start and end position (x, y) of each track was extracted from the results of TrackMate analysis. Then, the direction of the track was categorised into four classes: 0°–90°, 90°–180°, 180°–270°, and 270°–360°, and plotted.

To check for calcium activity, histogram analysis was conducted. An average of the Z-stack images was obtained from the treated images as above, the marginal zone was divided into four regions (s1–s4; starting from the edge), and histogram analysis conducted in Fiji. The results were plotted, and non-linear curve fitting (LogNormal function) was conducted to get a single line graph.

### Immunohistochemistry

Embryos were fixed for 1 h with 4% paraformaldehyde in calcium-magnesium free phosphate buffered saline (PBS, pH7.4), dehydrated with MetOH, and rehydrated stepwise with serial dilutions of PBS containing 1% triton X-100 (PBST). The samples were then treated in ice-cold acetone at −20 °C for 20 min. After washing in PBST, they were blocked with blocking buffer (PBST containing 5% normal goat serum and 0.02% thimerosal) for 2–6 h at room temperature on a rocker. Embryos were then incubated at 4 °C for 2–3 days with primary antibodies diluted 1:400: phospho-Smad1/5/8 (Cell Signaling Technology, 13820), NF-κB p65 (Abcam, ab16502)[71], NFAT4/NFATC3 (LSBio, LS-B6971). After washing with PBST, the embryos were incubated at 4 °C for 1 day with Alexa Fluor 488-conjugated goat anti-rabbit IgG (A11008, Invitrogen) diluted 1:200. For nuclear staining, 2.5 μg/ml of 4′,6-dia-midino-2-phenylindole was applied to the embryos or sections for 10 min and washed thoroughly. After mounting on a slide, the stained embryos were imaged with a Leica SPE1 confocal microscope. The images were processed in using Fiji to generate a maximum projection.

### Modelling

We model a 1-dimensional ring of 100 cells ($i = 0,...,99$), with periodic boundary conditions, representing the marginal zone. Each cell being modelled includes interactions between three key substances during primitive streak formation: a streak inducer, cVG1/GDF3 ($V_i$), a streak inhibitor, BMP4 ($B_i$) and calcium activity, ($C_i$). In addition, we model that each cell can either participate in streak formation, or not. We label this binary state as committing to 'streak identity' and assume it is analogous to expression of NODAL, downstream of cVG1.

We model cVG1/GDF3 and BMP4 as the concentrations of these proteins external to each cell and as functions of their production and decay. These are secreted proteins, and thus we assume that their propagation from cell to cell involves secretion by one cell, sensing by another cell, followed by activation of the production of the same ligand by the receiving cell. In contrast, calcium activity can be transmitted between neighbouring connected cells through gap junctions and its propagation is therefore faster. We model calcium activity, as a proxy for the combination of high amplitude of calcium firing along with the local proportion of firing cells at any one time point. As the marginal zone of the real embryo is several cells wide, whereas the model represents it as one-element wide, our model compresses the local state at a particular level down to that single element.

The model is initiated with no cells showing 'streak identity' and with zero cVG1 and calcium activity in all cells. We model a shallow, linear gradient of BMP4 highest in the anterior marginal zone ($B_{0,99} = 1.1$ and $B_{49,50} = 2.2$), mimicking results of in situ hybridisation and RNAseq experiments in the pre-streak embryo[14]. We assume that cVG1 and BMP4 are mutually antagonistic, due to their activation of Smads 2/3 and 1/5/8 respectively, which then compete to bind with Smad 4 intracellularly[72]. Mathematically, we encode this relationship by assuming that:

- When BMP4 drops below a first threshold, cVG1 is induced.
- cVG1 increases the decay rate of BMP4.

The initial gradient of BMP4 implies that the level of BMP4 in the posterior marginal zone is sufficiently low ($B_i < \beta_V$) for cVG1 to be expressed in these cells. Therefore, within the first 0.5 h of the simulation, the posterior marginal zone displays a region of cVG1 expression also mimicking the results of in situ hybridisation and RNAseq experiments in the pre-streak embryo.

Initially, no cell has become committed to 'streak' identity and, in the absence of calcium activity, the level of BMP4 decays linearly. And:

- When BMP4 drops below a second threshold, cells commit to a streak identity.
- Cells with streak identity initiate calcium activity, which then propagates between cells.

We then assume that calcium activity has a dual effect on the level of BMP4:

- Calcium activity above a very low threshold induces BMP4.
- However, calcium activity promotes decay of BMP4.

Together these assumptions imply that, in an intact embryo, low levels of BMP4 initiate the formation of a streak. The presence of a streak adjacent to the posterior marginal zone stimulates a gradient of calcium activity, highest posteriorly, which strengthens the BMP4 gradient.

When the site of streak formation is removed in an isolated anterior half of an embryo, calcium activity dies down, in turn resulting in the decay of BMP4. cVG1 is initiated on both sides of the embryo transiently. BMP4 continues to decay until a streak is initiated on one side. The initiation of a streak on one side, from which calcium activity propagates across the embryo, prevents the formation of a streak at the opposite side of the fragment.

In addition, we assume that calcium activity, cVG1 and BMP4 decay linearly. The interactions and model assumptions listed can be encoded by the following equations. We use H(x) to denote the Heaviside function, which we define explicitly. The interactions modelled are also displayed graphically in Fig. 6a:

$$\frac{dB_i}{dt} = k_B H(C_i - \alpha) - (\gamma_0 + \gamma_C C_i + \gamma_V V_i) B_i,$$
$$H(C_i - \alpha) = \begin{cases} 1, \text{where } C_i - \alpha \geq 0 \\ 0, \text{where } C_i - \alpha < 0 \end{cases} \quad (1)$$

$$\frac{dC_i}{dt} = k_C A_i + \frac{D}{\Delta x^2}(C_{i+1} + C_{i-1} - 2C_i) - \lambda C_i,$$
$$A_i = \begin{cases} 1, \text{where } \beta_C - B_i(\tau) \geq 0 \text{ for any } \tau < t \\ 0, \text{where } \beta_C - B_i(\tau) < 0 \text{ for all } \tau < t \end{cases} \quad (2)$$

$$\frac{dV_i}{dt} = k_V H(\beta_V - B_i) - \mu V_i,$$
$$H(\beta_V - B_i) = \begin{cases} 1, \text{where } \beta_V - B_i \geq 0 \\ 0, \text{where } \beta_V - B_i < 0 \\ \beta_V > \beta_C \end{cases} \quad (3)$$

All parameter values are given in Supplementary Table 2; the effects of using Hill functions (with a Hill coefficient of 4) for the production of BMP4 and for the production of cVg1 instead of Heaviside functions was also explored; the results are described in Supplementary Information and shown in Supplementary Movies 8 and 9. This system of equations is solved using the Euler method with $\Delta t = 0.0001$ h, implemented in Python (https://github.com/catohaste/multiple-streak-inhibition).

## Live imaging of Ca²⁺ activity in microfluidic human amniotic sac embryoids

All protocols used in the work with hPSCs to model the development of embryonic-like sacs and primitive streak-like cells have been approved by the Human Pluripotent Stem Cell Research Oversight Committee at the University of Michigan, Ann Arbor. The microfluidic amniotic sac embryoid (μPASE) assay was performed as previously described[43]. Briefly, the microfluidic device is fabricated using a standard soft lithography technique. The device consists of three parallel channels. Geltrex (Thermo Fisher) is diluted to 70% using E6 medium and loaded into the central gel channel separated from the side channels by trapezoid-shaped supporting posts. Upon gelation, the Geltrex matrix generates concave pockets between supporting posts for cell seeding. hESCs suspended in mTeSR1 medium were introduced into the cell loading channel and allowed to settle and cluster in the gel pockets. After 24 h of cell seeding, mTeSR1 medium was replaced by a basal medium (E6 and 20 ng/ml bFGF) containing 10 μM fluorogenic calcium-sensitive dye (Cal-520, Abcam). In total, 50 ng/ml BMP4 was supplemented into the cell seeding channel to generate μPASE, wherein hESCs facing directly towards the BMP4 stimulation side differentiated into squamous amniotic ectoderm-like cells; hESCs at the opposite pole differentiated into primitive streak-like cells. If no BMP4 was supplemented into the device (control), by 24 h, the hESC cluster would develop into a columnar cyst with a central lumen, and remain pluripotent. The microfluidic device was then placed in an incubator. After 24 h incubation, the medium was replaced by fresh basal medium without Cal-520, and then placed on an Olympus DSUIX81 spinning-disc confocal microscope equipped with an EMCCD camera (iXon X3, Andor). Images were captured every two seconds for 12 min. The images were processed to correct for fluorescence bleaching using the simple ratio method in Fiji.

## Assay with human gastruloids in suspension culture

All work with human embryonic stem cell lines performed in the Crick Institute has been carried out with approval from the UK Stem Cell Bank Steering Committee and adheres to the regulations of the UK Code of Practice for Use of Human Stem Cell Lines. The research on gastruloids is compliant with the ISSCR 2021 Guidelines for Stem Cell Research. Human embryoids were established as previously described[73]. Briefly, single cells of hESCs from the RUES2-GLR line[74] were resuspended in NutriStem hPSC XF Medium (Biological Industries 05-100-1A) with 5 μM ROCK inhibitor (Y-27632; Bio-Techne 1254/10). Six-well plates were seeded with 40,000–50,000 cells/well, and the cells were cultured in NutriStem XF medium for 4 days, to reach ~60–80% confluence. The cells were pre-treated with 3.25 μM Chiron in NutriStem XF on the fifth day and dissociated in 0.5 mM EDTA for 5 min, followed by gentle trituration. Single cells were centrifuged and resuspended in Essential 6 (E6, Gibco A15164-01) aggregation medium containing 5 μM ROCK inhibitor. A plating suspension was prepared from E6 medium with 5 μM ROCK inhibitor and 0.5 μM Chiron (CHIR 99021, Bio-Techne 4423/10), with a density of 10 live cells/μl. In total, 40 μl of plating suspension was transferred to each well of non-cell-adhesive sterile round-bottomed 96-well microplates (Greiner 650970) as required. Plates were centrifuged at 70 × g for 2 min, followed by 170 × g for 1 min to generate a pellet of cells. A single aggregate formed in each well within 3 h of plating, from a starting number of ~400 live cells. For chemical treatment 24 h after plating, U-73122 hydrate (Merck Sigma-Aldrich U6756) or U-73343 (Merck Sigma-Aldrich U6881) reconstituted in DMSO (Merck Sigma-Aldrich D2438) at double concentration was added as 40 μl volumes to the 40 μl aggregation volume in each well. Embryoids were imaged by confocal microscopy using a Molecular Devices ImageXpress Confocal HT.ai with a 60 μm pinhole spinning disc module and an ANDOR Zyla 4.2 sCMOS camera with 2 × 2 pixel binning. The BRA-mCerulean, SOX2-mCitrine and SOX17-tdTomato fluorophores were excited with a LED laser light source (89 North LDI-NIR-EX) and 445/20 nm, 520/10 nm and 555 nm excitation filters, respectively. Emitted light was collected through the following emission filters (Semrock): FF01-483/32, FF01-562/40 and FF01-624/40, respectively. Images were collected as z-stacks (108 μm in 12 μm steps) and were recorded as maximum intensity projections for fluorescence channels and Best Focus images under phase contrast. Imaging data were recorded with MetaExpress software (6.7.1.157). Images were adjusted for contrast in ImageJ2[70] by changing the minimum and maximum displayed pixel values for each fluorescence channel, using consistent levels across the plate to allow comparison of different experimental conditions.

## Assay with *Drosophila* embryos

Two hours after egg deposition, Oregon R *Drosophila* embryos were collected and dechorionated in a 50% bleach solution before being incubated with either U73122 (20 μM) or U73343 (20 μM) for 30 min.

Embryos were left to age for an additional hour in Schneider's Insect Medium (Gibco) before fixation using a standard paraformaldehyde, heptane, and methanol procedure[75]. Fixed embryos were then mounted using Slowfade Diamond with DAPI (Thermo Fisher Scientific). Embryos were visualised using an Olympus FV3000 and gastrulation assessed by key markers of the onset of gastrulation, marking primarily the formation of the ventral furrow and posterior midgut invagination.

## Reporting summary

Further information on research design is available in the Nature Portfolio Reporting Summary linked to this article.

## Data availability

All data generated in this study are provided in the Supplementary Information and accompanying Source Data file. Source data are provided with this paper.

## Code availability

The simulation code has been deposited in a persistent GitHub repository https://github.com/catohaste/multiple-streak-inhibition (https://doi.org/10.5281/zenodo.10523300).

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

## Acknowledgements

We are grateful to Sandip Patel, Anant Parekh, Andrea Streit and Octavian Voiculescu for advice on experiments and helpful comments on the manuscript. We also acknowledge the Cell Services Science Technology Platform at the Francis Crick Institute for providing regular Mycoplasma screening of the hESC line, the Zoology Imaging Facility of Cambridge University for assistance and support with microscopy on *Drosophila* embryos, and Sophie Brumm for providing a pSMAD staining protocol. This study was funded by a Wellcome Trust Investigator Award (107055/Z/15/Z) to C.D.S., which supported H.C.L., H.-C.L. and N.M.M.O. H.C.L. was also supported by a fellowship from the Basic Science Research Program through the National Research Foundation of Korea (NRF) (2014R1A6A3A03053468). N.M.M.O. was also funded by UCL. C.H. was funded by a Medical Research Council Doctoral Training Programme studentship (MR/N013867/1). P.B.-B. and N.M. were funded by the Francis Crick Institute which receives its core funding from Cancer

Research UK (CC2186), the UK Medical Research Council (CC2186), and the Wellcome Trust (CC2186). A.A.M. was funded by a studentship from the Anatomical Society of Great Britain and Ireland in C.D.S.'s lab. Y.Z. and J.F. were funded by the Michigan-Cambridge Research Initiative, the US National Institutes of Health (R21 HD100931), and the National Science Foundation (CMMI 1917304). E.L.W. was funded by a BBSRC DTP scholarship. T.T.W. was funded by the University of Cambridge ISSF (097814) and the Wellcome Trust (200734/Z/16/Z).

## Author contributions

Conceptualisation: H.C.L. and C.D.S.; Methodology: all authors; Software: C.H., H.-C.L. and A.A.M.; Investigation, analysis and validation: H.C.L., C.H., N.M.M.O., P.B.-B., A.A.M., H.-C.L., Y.Z. and E.L.W.; Writing: all authors; Supervision: C.D.S., T.T.W., K.M.P., J.F. and N.M.; Project administration: C.D.S.; Funding acquisition: C.D.S., T.T.W., J.F. and N.M.

## Competing interests

The authors declare no competing interests.
