## [Peer Review File · Nature Communications]

Regulation of long-range BMP gradients and embryonic polarity by propagation of local Calcium-firing activityREVIEWER COMMENTS

Reviewer #1 (Remarks to the Author):

It has been established that the formation of the primitive streak of the avian embryo involves antagonistic interactions between inductive signals (GDF3) in the posterior and repressive signals (BMP) in the anterior. The way these signals interact to result in the robust formation of only a single primitive streak remains to be elucidated. The experiments in this paper address the how and where of potential interactions. The first set of observations concerns the route of signalling. Using a combination of small lesion experiments it is concluded that a key signal travels predominantly through the marginal zone, i.e. the border between the embryonic and extra-embryonic area. Furthermore, it is suggested that a critical component of the signal travels through gap junctions and therefore is likely a small molecule (<1000 dalton). Experiments are then presented that suggest that this signal could be calcium, based on observation of calcium signalling dynamics and a number of perturbation experiments. The authors then continue to investigate possible connections between calcium and BMP signalling and propose that NFkB and NFAT are proposed to be key mediators. The observations show a region of high calcium transients initiated in the posterior marginal zone and slowly progressing towards the anterior. It is suggested that the gradient of this calcium activity interacts with and regulates the opposing BMP gradient. This is thought to be mediated through differential effects of calcium levels and transients on NFkB and NFAT activity that have opposing effects on BMP activity. High calcium spikes favour high NFkB levels which inhibits BMP signalling and low sustained calcium levels favour high NFAT levels that result in increased BMP activity and oppose streak formation.

A basic model of these interactions is shown to be capable of recapitulating some key characteristics of this signalling system. Finally, it is argued on the basis of some preliminary data and existing insights from the literature that this signalling pathway uncovered here involved in axis formation is likely conserved from insects to humans.

This study reports a number of interesting, novel and tantalizing observations, involving a number of potential new players in the interactions between streak inducing and inhibiting signals, but also raises/leaves a number of points that need to be addressed.

A causative role for calcium is very difficult to show conclusively since it has many pleiotropic effects. The fact that flooding an area of cells with calcium, through the use of an ionophore incapacitates these cells (mimicking deletion) is hardly surprising but does not show that calcium is an instructive signal. To demonstrate that the phospholipase C inhibitor results, which is shown to increase TBXT and decrease GATA 2 expression act through effects on calcium, it will be necessary to show how calcium levels and/or transients are affected by this inhibitor, locally and globally. Other experiments manipulating/clamping calcium levels and transients at defined levels could strengthen the argument for causation, rather than being a secondary cellular response.

This is linked to the question of why the gradient in calcium activity progresses so slowly. It would suggest that the reported small scale wave propagation does not play a major role in streak formation. If it does can this be further elaborated?

Assuming that the calcium activity gradient is an instructive signal, mediating its effect through the action of NFkB and NFAT as proposed, can it be shown that the calcium gradient development is reflected in a gradient of NFkB and NFAT activation? This could be done through for instance measurements of nuclear localisation in the MZ at different times or following expression of expression of certified target genes. The data in Fig S6 h showing localisation of NFkB and NFAT are not very informative and leave me wondering whether the antibodies used are specific.

The model assumes that cells with streak identity produce calcium, can it be shown that ectopic GDF3 expression can activate calcium signalling and NFkB activation?

Minor comments

It would appear from Fig. 2A that calcium signalling is not necessarily confined to the marginal zone, but entails the whole area opaca. This would be better in line with the expression pattern of the gapjunction components (FigS2e). It would be desirable to show whole embryo overview images at various time points to understand better where the enhanced calcium signalling takes place and how it develops over time also applies to Fig S4 a,b,c.

Fig. 1. According to the figure the number of embryos that form a streak is always 100%. This would be very unexpected since in any experiments a number of embryos will not develop until the streak stage, regardless of any perturbations. I assume that the graph reflects only the embryos that actually formed a streak (started to express TBXT), this should be clarified in the legend and the methods. It would be better to report the actual success rate in the graphs through the paper.

Also in fig S6,c,d,f it shows numbers like 9/9 what does this mean? Does it imply that the pattern is identical for 9 out of 9 embryos, some better form of quantification would be useful.

From the calcium movies provided and Fig. S3 it is not clear that the signal actually propagates significantly beyond neighbouring cells. It appears that most of the temporal variation in brightness is localised to specific cells. The methods section describes that the analysis was performed using a software package designed to track Ca^{++} signal propagation in neuronal networks. Could the assumption of propagation have biased the results? If this is to be maintained then it would be desirable to show a movie showing propagating calcium signals overlaid with the derived propagation vectors.

Describe variables in the model section better, which are Ca^{+} , BMP Vg1, streak/nodal_on etc. The two-threshold assumption is a very strong one. How necessary is this to obtain the results? How were the parameters estimated to achieve a time scales? The model assumes that the initial condition is a gradient of BMP. This does not seem unreasonable but should be explained better.

Reviewer #2 (Remarks to the Author):

The manuscript by Lee et al addresses a critical process in animal development – the primitive streak (PS) formation. The authors show here that the position of the PS is specified through an interplay between BMP and Ca^{2+} and that this interplay is necessary to ensure that only one PS is specified in an embryo. More specifically, based on a combination of embryological, pharmacological and mathematical modelling experiments, they conclude that a wave of Ca^{2+} activity originating at the PS in chick embryos organizes an anterior-posterior BMP activity gradient, and this interaction prevents the generation of additional primitive streaks in the embryo. The authors then go on to demonstrate that this mechanism is likely conserved in many species, both invertebrates (*Drosophila*) and vertebrates (human embryoids/gastruloids). This is a very interesting study and would be a valuable addition to the field of developmental biology. However, in order for it to be considered for publication, the following comments need to be addressed.

1. Ca^{2+} wave: In the movies provided, it is difficult to appreciate an obvious wave of Ca^{2+} activity. The authors say that, “ Ca^{2+} wave analysis was conducted separately for each selected region (area opaca, marginal zone or area pellucida) and was performed at the single-cell level (which involved identifying a fluorescent cell in one frame, then another firing cell within a certain distance to the first cell in the next frame, and linking them), using the TrackMate plugin in Fiji.”

In this context, I have a few questions.

First, what was the distance used as the radius from the first firing cell to identify a track. It would be

important to specify here so as to be able to compare it with the size of single cells.

Second, considering that the tracks identify change direction very frequently, how do the authors know if a track they identify is indeed 'one track'. For that matter, how do they know if there is a wave of Ca²⁺ activity rather than just random firing of Ca²⁺ in different cells.

Third, some cells fire repeatedly. Considering that the authors propose that cells at a distance fire in a synchronous manner due to cytonemes connecting them, do they see such groups of cells repeatedly firing together? How prevalent are these connections in the chick embryo? Furthermore, although it is not necessary in the context of this study, could these connections be 'ablated' to test whether that is how pairs of cells at a distance fire together?

Finally, in movie 2, many of the cells firing seem to be dividing at the same time. Ca²⁺ activity has been shown previously to be high during cell divisions. What is the contribution of cell divisions to cell firing in the relevant time periods?

2. On lines 200-202, the authors state that, "Grafting a DM-bead increases the incidence of Ca²⁺-active cells near the bead, along with reduced frequency and increased amplitude of Ca²⁺-oscillations relative to controls grafted with a DMSO-bead (Fig. 4, a and b)."

However, in the graph in fig. 4B(i), the higher incidence of Ca²⁺ active cells cannot be seen. In two (e1 and e3) out of three experiments, the incidences seem to be very similar to the control group. If this point has to be made in the study, further repeats and statistics need to be included for this analysis.

3. Fig. 1d requires a multiple comparisons correction in addition to Boschloo's test as four groups are being compared. If such a test was already performed, a mention of the same was missed in the figure legends.

4. Lines 140-142: "MZ cells fire less frequently (average frequency=4.90±0.24 mHz in the posterior MZ and 5.26±0.41 mHz in the posterior AP) but with greater amplitude (1.3-fold) than cells of the AP (Fig. 2d)."

From the boxplot in Fig. 2d(ii), the distributions (pAP and pMZ) look very similar. In the absence of a statistical test, this statement is unsubstantiated.

5. Lines 325-330: "To study the role of Ca²⁺ signalling in *Drosophila* gastrulation, we treated early *Drosophila* embryos with U73122: this caused a significant (P = 0.0375; Boschloo's test, n=56) delay in the initiation of gastrulation compared to control embryos treated with U73343 (n=34) suggesting that Ca²⁺ activity may also be involved in regulating the initiation of gastrulation in the fly (Extended Data Fig. 8c)."

Is only the onset of gastrulation is delayed? Or other processes also delayed/slower. Normalization is required in order to assess if the effect is on gastrulation onset or on development in general.

Reviewer #3 (Remarks to the Author):

General Comments:

The authors of this paper have employed a combination of computational modeling and experiments to investigate a potential mechanism for the formation of long-range molecular gradients that regulate embryonic polarity during primitive streak formation in early embryogenesis. They observed and quantified transient Ca^{2+} activity spikes, which resulted in signaling waves along the peripheral extraembryonic marginal zone in chick embryos. Through computer simulations, the authors suggest that Ca^{2+} dynamics play a role in regulating embryonic polarity by interacting with the established BMP gradient. While the study is potentially interesting, there are several areas that require improvement, including data analysis, the proposed mathematical model and execution, and comparison between experiments and simulations.

I would like to start by acknowledging the amount of work dedicated to this study and congratulating all the authors for their valuable contribution to this manuscript. The paper is highly readable and engaging, with well-structured figures. The observation and quantification of calcium waves during primitive streak formation are particularly intriguing.

As a reviewer with expertise in mathematical and computational biology, my main concern lies in the mechanistic interpretation of the effect of calcium waves on pre-established morphogenetic signals. The authors have developed a mathematical model to guide the interpretation of the proposed mechanism. Consequently, my review will focus more on this aspect of the manuscript. Several additional steps need to be addressed to ensure that the model comprehensively supports the experimental observations.

1. Complete definition of the model:

- The model equations and boundary conditions require a more thorough explanation. The current presentation lacks both mathematical and descriptive details regarding the boundary conditions.
- The initial conditions should be clearly specified both mathematically and in the text. Additionally, the authors should provide a well-justified rationale for their choice of initial conditions. If the model solutions are sensitive to initial conditions, conducting a scan with different initial conditions would be beneficial. It is acceptable to describe the model in the main text if necessary, but keeping all model details in the methods section would provide greater clarity.
- Variables and parameters need explicit descriptions, including the provision of units. While the work adequately describes the variables, only a general parameter type description is given in Extended Data Table 2. Providing units for all variables and parameters is essential. Currently, units are only provided for time (in hours) when describing the time discretization step.

2. Phase space of model solutions:

- To validate the model, it is crucial to explore the phase space of its solutions thoroughly. Conducting a comprehensive parameter scan will reveal the different solutions that the model can produce. This analysis will help identify parameter ranges that align with experimental observations and provide a set of potential perturbations for experimental validation.
- The authors have only provided one parameter set without justifying their choice. Given the use of parameter-dependent Heaviside functions in the model, variations in the parameter set will strongly influence the model solutions and their fit with experimental observations.

3. Clarification and modification of the model:

- The transition from calcium waves to a gradient needs to be rationalized, derived, and quantified in terms of gradient formation dynamics, length-scale, time-scale, and the transport mechanism regulating gradient formation. The authors should reference relevant literature on the stability of pulse and traveling waves, such as the fire-diffuse-fire model, to support their proposed hypothesis.
- Currently, calcium is modeled as a diffusive molecule using a reaction-diffusion equation that does not allow for solutions in the form of traveling waves. The authors should provide a clear justification for this modeling choice.
- Morphogens such as BMP and VG1 are modeled as ordinary differential equations (ODEs) without justification for their homogeneous description.

In conclusion, this work proposes a mechanism for modulating the shape and dynamics of morphogen gradients through the rapid spread of calcium waves. The authors have gathered sufficient experimental data to feed a robust mechanistic model that could strongly support their findings and hypothesis. As a suggestion, the authors could develop a self-consistent model that proposes a mechanism to explain these observations without relying on branched equations. The presented elements and interactions in the model suggest that patterning and the robust establishment of the primitive streak can be self-organized through the interplay of an activator-inhibitor structure and rapid traveling wave spread.

As minor comments:

1. The GitHub link provided in the manuscript does not work. It would be helpful for the authors to review the code and provide further explanation and comments.

2. The authors state that "morphogen gradients can reliably be generated by diffusion only over a limited range (<100 cell diameters, or 1mm)." However, there are examples of robust long-range morphogen gradients, such as in planaria (DOI: 10.1016/j.devcel.2016.12.024, 10.1088/1478-3975/ac86b4). The authors should consider this literature and provide a discussion of how their findings align or differ from those studies.

3. The authors mention that "The diameter of the area pellucida and MZ (excluding the outer extraembryonic area opaca) in pre-primitive-streak stage embryos has an average diameter of 240 cells; therefore, the half-circumference of the MZ would span approximately 380 cell lengths, suggesting that the signal travels faster than 1 cell/min (≥ 1.27 cells/min). This fast transmission rate suggests an intercellular mechanism based on small molecules, such as ion currents." This speed corresponds to an effective diffusion coefficient on the order of $2 \mu\text{m}^2/\text{s}$, which can also fit with a diffusion-based morphogen spread.

4. The authors mention that "Misexpression of either GJB2 or GJB6 rescued the marginal-zone-excision phenotype, decreasing the frequency of ectopic primitive streak formation relative to both a control plasmid and to misexpression of GJA5 (Fig. 1, d-e), which should not form functional connections with GJBs (Extended Data Fig. 2, f-g)." If signal spread is a result of morphogen diffusion, altering junctions can change the transport dynamics of such a morphogen, resulting in slower gradient formation and a shorter-range length-scale. Have the authors extended the timescale of this experiment to observe delayed patterning? Additionally, short-range gradients resulting from altered junctions can potentially explain changes in the likelihood of primitive streak formation.

5. The authors mention that "We explored this further using mosaic electroporation of an expression vector encoding the Ca^{2+} -reporter GCaMP6; high-resolution imaging revealed very thin intercellular processes connecting firing cells that were not adjacent (Fig. 2 bi-biv and Extended Data Movie 4). These processes are reminiscent of the cytonemes described in *Drosophila*¹⁹ and other systems, including the somite in early chick embryos²⁰. These observations suggest that Ca^{2+} spikes may be transmitted via cytoneme-like processes (Extended Data Movie 4)." However, the notion of signal transport via cytonemes is still controversial and lacks clear agreement in *Drosophila*. Evidence for signal transport via cytonemes in chick embryos is limited, and the authors need to conduct a thorough study to support this claim. I suggest that the authors either provide stronger evidence for this type of transport mechanism or temper their language and suggest complementary and alternative transport possibilities.

6. The authors state that "The above results suggest that relatively short-range transmission of Ca^{2+} transients conveys positional information over a large embryonic territory to regulate the site of primitive streak formation." It would be helpful to clarify the length of the range referred to in this statement. Previously, it was mentioned that Ca^{2+} waves can span 200 μm , which is not considered

short-range in the context of information spread. It may be beneficial to provide this distance in the context of other long-range signals in this system.

7. When the authors propose that "we hypothesized that a gradient of Ca^{2+} activity, rather than Ca^{2+} concentration, along the MZ might act as the positional cue," please clarify the difference between Ca^{2+} activity and Ca^{2+} concentration. Additionally, as mentioned earlier, this hypothesis requires more background information. The authors have described the observation of traveling Ca^{2+} waves, but transitioning from traveling waves to stable gradient formation necessitates further reasoning before proposing this hypothesis.

Overall, the paper has great potential; however, improvements are necessary in the mathematical model, and the mechanistic interpretation to support the conclusions. I encourage the authors to address the mentioned concerns for a more robust and comprehensive understanding of the experimental observations.

Dr. Daniel Aguilar-Hidalgo

Response to reviewer's comments

We are grateful to all the reviewers for their extremely helpful and constructive comments. We have followed all their suggestions and comments and have made many changes to the manuscript and figures, including some additional work. We feel that the manuscript has improved very substantially as a result of this revision and we greatly appreciate the professional attitude and helpfulness of the reviewers and the editor in helping us to improve this paper. Below we address the comments of each reviewer and provide brief summaries of what we have done, and answers to their questions.

Reviewer #1 (Remarks to the Author):

It has been established that the formation of the primitive streak of the avian embryo involves antagonistic interactions between inductive signals (GDF3) in the posterior and repressive signals (BMP) in the anterior. The way these signals interact to result in the robust formation of only a single primitive streak remains to be elucidated. The experiments in this paper address the how and where of potential interactions. The first set of observations concerns the route of signalling. Using a combination of small lesion experiments it is concluded that a key signal travels predominantly through the marginal zone, i.e. the border between the embryonic and extra-embryonic area. Furthermore, it is suggested that a critical component of the signal travels through gap junctions and therefore is likely a small molecule (<1000 dalton). Experiments are then presented that suggest that this signal could be calcium, based on observation of calcium signalling dynamics and a number of perturbation experiments. The authors then continue to investigate possible connections between calcium and BMP signalling and propose that NFκB and NFAT are proposed to be key mediators. The observations show a region of high calcium transients initiated in the posterior marginal zone and slowly progressing towards the anterior. It is suggested that the gradient of this calcium activity interacts with and regulates the opposing BMP gradient. This is thought to be mediated through differential effects of calcium levels and transients on NFκB and NFAT activity that have opposing effects on BMP activity. High calcium spikes favour high NFκB levels which inhibits BMP signalling and low sustained calcium levels favour high NFAT levels that result in increased BMP activity and oppose streak formation.

A basic model of these interactions is shown to be capable of recapitulating some key characteristics of this signalling system. Finally, it is argued on the basis of some preliminary data and existing insights from the literature that this signalling pathway uncovered here involved in axis formation is likely conserved from insects to humans.

This study reports a number of interesting, novel and tantalizing observations, involving a number of potential new players in the interactions between streak inducing and inhibiting signals, but also raises/leaves a number of points that need to be addressed.

A causative role for calcium is very difficult to show conclusively since it has many pleiotropic effects. The fact that flooding an area of cells with calcium, through the use of an ionophore incapacitates these cells (mimicking deletion) is hardly surprising but does not show that calcium is an instructive signal. To demonstrate that the phospholipase C inhibitor results, which is shown to increase TBXT and decrease GATA 2 expression act through effects on calcium, it will be necessary to show how calcium levels and/or transients are affected by this inhibitor, locally and globally. Other experiments manipulating/clamping calcium levels and transients at defined levels could strengthen the argument for causation, rather than being a secondary cellular response.

We thank the reviewer for this comment. Indeed, we have explored several chemicals that can modulate intracellular calcium for their effect on Ca²⁺ activity in the marginal zone. For this, it is critical to apply the chemicals to a small region of the marginal zone, so we tried to use affinity or ion exchange beads for their delivery. We tested: U73122, 2-APB, forskolin, nicardipine, and RP-cAMP, using either Affi-Gel Blue or AG1X2-formate beads. Unfortunately, none of these treatments had any effect on Ca²⁺ activity compared to controls using the beads alone. As we are limited by current technology, we were unable

to find a method to answer this question. Despite this, we believe that we have shown strong correlation between the Ca^{2+} firing activity (specifically the frequency and amplitude of the pulses, as well as the number of firing cells) with embryo polarity; therefore if this activity is not itself directly responsible, it is most likely to be an event that is directly associated with it, such as the IP3/DAG system.

This is linked to the question of why the gradient in calcium activity progresses so slowly. It would suggest that the reported small scale wave propagation does not play a major role in streak formation. If it does can this be further elaborated?

As the reviewer correctly notes, we observe progression and accumulation of short-range Ca^{2+} waves rather than long-range calcium waves. Not all cells in the marginal zone seem to fire, but only a subset. We are proposing a model in which Ca^{2+} activity in a cell represses BMP activity (as proposed in *Drosophila* for the regulation of Dpp and dorsoventral polarity; [Wharton et al. 1993; Markova et al., 2015]). As BMP is under positive feedback regulation (e.g.: Jones et al., 1992; Metz et al., 1998; Re'em-Kalma et al., 1995; Schulte-Merker et al., 1997; Rentsch et al., 2005), regions of high BMP activity (i.e. low Ca^{2+} activity) should result in increased BMP production, which will then signal to neighbouring cells and amplify the effect, as well as smooth out the BMP concentration among cell neighbours regardless of the proportion of those cells that are firing. This mechanism is consistent with the observed speed of propagation of the activity wave front across the diameter of the embryo, as well as with the progression of positional information (as revealed by the sequential ablation experiment). In retrospect, thinking about the reviewer's comment, we realise that we had not explained this clearly enough in the original manuscript. We have therefore altered the text accordingly in a number of places, including a slight change in the title of the paper, to avoid confusion.

Assuming that the calcium activity gradient is an instructive signal, mediating its effect through the action of NFkB and NFAT as proposed, can it be shown that the calcium gradient development is reflected in a gradient of NFkB and NFAT activation? This could be done through for instance measurements of nuclear localisation in the MZ at different times or following expression of expression of certified target genes. The data in Fig S6 h showing localisation of NFkB and NFAT are not very informative and leave me wondering whether the antibodies used are specific.

As the reviewer commented, we expected to see a spatial gradient of NFkB (and/or NFAT) expression as shown in the dorso-ventral axis of fly embryos. However, we could only observe local differences compared to control (as shown in new Supplementary Fig. 7h) but not a long-range gradient using our current techniques. One of several possible explanations is that in the chick marginal zone not all cells are undergoing Ca^{2+} firing at a given time (see above); their non-firing neighbours could be regulating their BMP status independently of NFkB/NFAT. Evidence for the specificity of the anti-NFkB antibody in chick embryos has been published previously (Li et al., 2017 – now added to the text).

The model assumes that cells with streak identity produce calcium, can it be shown that ectopic GDF3 expression can activate calcium signalling and NFkB activation?

Thank you for suggesting this interesting experiment. At the moment, GDF3/Vg1 is not available as a pure protein. Instead, to address this reviewer's comment, we have now used the related protein ACTIVIN, which mimics the activity of GDF3/Vg1 (Lee & Hastings et al., 2022), and checked changes in Ca^{2+} activity (new Supplementary Fig. 5 g). The data show that like Dorsomorphin, ACTIVIN treatment also increases the number of firing cells compared to controls. However, the amplitude of the Ca^{2+} activity was lower than in controls and there was no difference in frequency – this is different from the results in embryos treated with dorsomorphin. For this reason, we propose that BMP inhibition has the key roles in communication across the embryo (rather than propagation of the “positive” inducing signal by GDF3/Vg1). We have added the new data in Supplementary Figure. 5 g. We thank the reviewer for suggesting this experiment, which we feel has further strengthened the manuscript.

Minor comments

It would appear from Fig. 2A that calcium signalling is not necessarily confined to the marginal zone, but entails the whole area opaca. This would be better in line with the expression pattern of the gap junction components (FigS2e). It would be desirable to show whole embryo overview images at various time points to understand better where the enhanced calcium signalling takes place and how it develops over time also applies to Fig S4 a,b,c.

We thank the review for this comment. In our data, calcium activity is enriched in the outer region of the embryo including marginal zone and area opaca, but more strongly in the marginal zone. Moreover, the expression of the type-B gap junction channels (GJB) encompasses both the area opaca and area pellucida. Nevertheless, two tissue excisions in the lateral area opaca did not induce an ectopic primitive streak (Fig. 1biv) (unlike the result of the same experiment in the marginal zone), suggesting that the main positional information for primitive streak formation flows through the marginal zone but not the area opaca. One possible reason for this is that the marginal zone has other functions that may cooperate with the calcium signalling – one possible candidate is ASTL, a metalloprotease of the Astacin family, which has roles in processing TGF β proteins – ASTL is confined to the marginal zone. It is possible that this activity is required to process GDF3/VG1, BMP, or both; there is also some evidence of cooperation between Tolloid-like Astacins and calcium (e.g. Ge et al., 2006 – DOI: 10.1074/jbc.M511111200; Canty et al., 2006 – DOI: 10.1074/jbc.M510483200).

Concerning the imaging of calcium activity across the entire embryo, the short duration of the transients and the large scale of the embryo limit the overall area that can be imaged at each time point to make a time-lapse movie. At most, 1/4 of the embryo's perimeter is the maximum we could monitor for each time point using the technology we had available. We could cover a larger area at lower magnification, but unfortunately this does not resolve individual cells and their activity sufficiently to be informative.

Fig. 1. According to the figure the number of embryos that form a streak is always 100%. This would be very unexpected since in any experiments a number of embryos will not develop until the streak stage, regardless of any perturbations. I assume that the graph reflects only the embryos that actually formed a streak (started to express TBXT), this should be clarified in the legend and the methods. It would be better to report the actual success rate in the graphs through the paper.

We thank the review for this comment. The number of embryos shown in the figure does include all the embryos used in each experiment and shows the actual rate of streak formation. We do normally discard any embryos that look unhealthy before using them for an experiment, and a rate of 100% primitive streak formation in normal or control-treated embryos is not unusual in our hands. To clarify this, we revised the methods section.

Also in fig S6,c,d,f it shows numbers like 9/9 what does this mean? Does it imply that the pattern is identical for 9 out of 9 embryos, some better form of quantification would be useful.

We have now explained this more clearly in the figure legends throughout the manuscript (Fig.4 and Supplementary Figs. 2 and 7). Briefly, "n/m" refers to the number of embryos that show the phenotype described (n), out of the total (m) in each case.

From the calcium movies provided and Fig. S3 it is not clear that the signal actually propagates significantly beyond neighbouring cells. It appears that most of the temporal variation in brightness is localised to specific cells. The methods section describes that the analysis was performed using a software package designed to track Ca⁺⁺ signal propagation in neuronal networks. Could the assumption of propagation have biased the results? If this is to be maintained then it would be desirable to show a movie showing propagating calcium signals overlaid with the derived propagation vectors.

We thank the review for this comment. To make this clearer, we have added more snapshots with magnified views showing propagation to Supplementary Fig. 3.

The software package for studying neuronal networks was used to find calcium firing cells. On the other hand, to study the propagation of Ca^{2+} firing among cells, the Trackmate plugin in ImageJ/FIJI was used. To clarify this, we have added an explanatory diagram of how we identify the propagating tracks of Ca^{2+} firing between cells (Supplementary Fig. 4). In terms of the direction of the Ca^{2+} trails, we investigated this by drawing vectors of each track; we find that the direction of propagation between nearby cells is random (Supplementary Fig. 5h).

As the reviewer commented, the identification of Ca^{2+} waves can depend on how we define them for analysis. To reduce possible bias, we considered different interpretations including cell movement, firing delay or cell division (see Methods). To make this clearer, we have added a schematic diagram of the methods as new Supplementary Fig. 4.

Describe variables in the model section better, which are Ca^+ , BMP Vg1, streak/nodal_on etc. The two-threshold assumption is a very strong one. How necessary is this to obtain the results? How were the parameters estimated to achieve a time scales? The model assumes that the initial condition is a gradient of BMP. This does not seem unreasonable but should be explained better.

Thank you for these questions. We have added a more detailed explanation of the variables (see also answers to Reviewer 3). Concerning the two thresholds, in part this was done to account for the observation that although Vg1 is expressed as a fairly wide crescent in the posterior marginal zone, only the centre of this domain lies next to the primitive streak forming region: our upper threshold determines the domain of expression of Vg1, whereas the lower threshold positions the primitive streak forming region. In the revised version, we have varied the values of these and other parameters to explore the robustness of the model as requested by Reviewer 3, which also addresses some of the points above.

Concerning the initial gradient of BMP, we have now made it more explicit that the initial conditions do include a BMP concentration gradient across the embryo; this fits with our observations that a BMP gradient (mRNA and target genes like GATA2) can be observed in normal embryos before Ca^{2+} firing becomes strong. We now also explain the model more clearly in terms of the cross-regulation between Ca^{2+} activity and BMP signalling throughout the manuscript (including the change of title; see above).

Reviewer #2 (Remarks to the Author):

The manuscript by Lee et al addresses a critical process in animal development – the primitive streak (PS) formation. The authors show here that the position of the PS is specified through an interplay between BMP and Ca^{2+} and that this interplay is necessary to ensure that only one PS is specified in an embryo. More specifically, based on a combination of embryological, pharmacological and mathematical modelling experiments, they conclude that a wave of Ca^{2+} activity originating at the PS in chick embryos organizes an anterior-posterior BMP activity gradient, and this interaction prevents the generation of additional primitive streaks in the embryo. The authors then go on to demonstrate that this mechanism is likely conserved in many species, both invertebrates (*Drosophila*) and vertebrates (human embryoids/gastruloids). This is a very interesting study and would be a valuable addition to the field of developmental biology. However, in order for it to be considered for publication, the following comments need to be addressed.

1. Ca^{2+} wave: In the movies provided, it is difficult to appreciate an obvious wave of Ca^{2+} activity. The authors say that, “ Ca^{2+} wave analysis was conducted separately for each selected region (area opaca, marginal zone or area pellucida) and was performed at the single-cell level (which involved identifying a fluorescent cell in one frame, then another firing cell within a certain distance to the first cell in the next frame, and linking them), using the TrackMate plugin in Fiji.”

In this context, I have a few questions.

First, what was the distance used as the radius from the first firing cell to identify a track. It would be important to specify here so as to be able to compare it with the size of single cells.

We thank the review for this comment. Because of this comment and similar ones from the other reviewers, we now realise that our original use of the word “waves” is misleading because it lends itself to the interpretation that there is a single wavefront of calcium activity that travels across the embryo. We have not observed this – rather, there is a progressive change in the level of activity of neighbouring cells, where the amplitude and the number connected firing cells changes over space and time and it is this that travels across the embryo rather than a wavefront of calcium concentration. We also observe short-range propagation of the calcium pulses though, and to clarify the difference we now refer to these as “tracks”, “short-range propagation”, “connected firing cells”, and other similar descriptions whenever it can lead to misinterpretation.

The minimum length of a Ca^{2+} track was set to 13 μm which is slightly larger than a single cell; this excludes the possibility that if a cell is moving while firing (between two adjacent frames – 3 seconds) this will be interpreted as a propagating track. The maximum length was set to 90 μm to be considered as a propagating track. When we extrapolate tracks, time-delay and cell division were also considered. We have provided more information of this in the methods section and now include a new Supplementary Fig.4, with better explanations in diagram form.

Second, considering that the tracks identify change direction very frequently, how do the authors know if a track they identify is indeed ‘one track’. For that matter, how do they know if there is a wave of Ca^{2+} activity rather than just random firing of Ca^{2+} in different cells.

As the reviewer notes, the identification of Ca^{2+} tracks can be dependent on how we define them using software during image analysis. To reduce possible bias in the analysis, we considered different conditions, such as cell movement, the sequence of firing between adjacent cells and cell division, and reflect them when we extrapolate Ca^{2+} waves (see Methods). While it is possible that on occasion we may misidentify two adjacent firing cells as a track, this would occur very infrequently and the pattern of propagation seen in the movies makes it clear that the vast majority do correspond to transmission of calcium activity (this should be obvious in some of the supplementary movies we include in the paper).

As mentioned above and also in our response to the other reviewers, we see a progression and accumulation of short Ca^{2+} tracks rather than large-scale calcium waves; in fact not all, but only a subset of cells in the marginal zone seem to fire at a given time. We propose that Ca^{2+} activity feeds back to regulate BMP activity, which will signal to neighbouring cells and affect Ca^{2+} activity again. This can explain how short waves, rather than one large wavefront, can transmit information over a long distance, in combination with BMP. We have tried to make this clearer throughout the revised manuscript, including a change of title.

Third, some cells fire repeatedly. Considering that the authors propose that cells at a distance fire in a synchronous manner due to cytonemes connecting them, do they see such groups of cells repeatedly firing together? How prevalent are these connections in the chick embryo? Furthermore, although it is not necessary in the context of this study, could these connections be ‘ablated’ to test whether that is how pairs of cells at a distance fire together?

We thank the review for this comment. As the reviewer commented, connected firing over cytoneme-like (filamentous) processes were observed repeatedly in some cells in our movies. Unfortunately, however, quantification of the cytoneme-like processes is difficult because these very thin cellular projections can only be seen if and when they are firing; we are therefore unable to determine how many such projections are present in the tissue, or their dimensions, or the proportion of such processes that contributes to propagation of the calcium signal.

Finally, in movie 2, many of the cells firing seem to be dividing at the same time. Ca^{2+} activity has been

shown previously to be high during cell divisions. What is the contribution of cell divisions to cell firing in the relevant time periods?

We have not checked thoroughly the rate of cell divisions. Some previous studies on cell division showed average 6-7 hours of cell cycle without big regional difference in chick embryos (Chuai et al., 2006) or blockade of cell division had no effect on streak formation (Cui et al., 2005).

2. On lines 200-202, the authors state that, "Grafting a DM-bead increases the incidence of Ca²⁺-active cells near the bead, along with reduced frequency and increased amplitude of Ca²⁺-oscillations relative to controls grafted with a DMSO-bead (Fig. 4, a and b)."

However, in the graph in fig. 4B(i), the higher incidence of Ca²⁺ active cells cannot be seen. In two (e1 and e3) out of three experiments, the incidences seem to be very similar to the control group. If this point has to be made in the study, further repeats and statistics need to be included for this analysis.

Indeed the reviewer is correct in this observation. To address this, we revised the sentence about the incidence of calcium active cells. While it is true that the effect of DM on the number of firing cells is only slight, there is much stronger (and significant) effect on the frequency and amplitude of the Ca²⁺ signal. We felt that we should still report the observation that we see a slight increase in the number of cells but now we are also pointing out that the effect is very small compared to amplitude and frequency.

3. Fig. 1d requires a multiple comparisons correction in addition to Boschloo's test as four groups are being compared. If such a test was already performed, a mention of the same was missed in the figure legends.

We thank the review for this comment. We have redone the statistics using a Chi-squared test, and we now show the statistical significance adjusted for multiple comparisons with the Holm-Bonferroni correction.

4. Lines 140-142: "MZ cells fire less frequently (average frequency=4.90±0.24 mHz in the posterior MZ and 5.26±0.41 mHz in the posterior AP) but with greater amplitude (1.3-fold) than cells of the AP (Fig. 2d)."

From the boxplot in Fig. 2d(ii), the distributions (pAP and pMZ) look very similar. In the absence of a statistical test, this statement is unsubstantiated.

Again, we agree with the reviewer. We now say that there is "no significant difference" in the frequency of firing between the two regions, but that the difference in amplitude is significant. Thank you for helping us to clarify this.

5. Lines 325-330: "To study the role of Ca²⁺ signalling in Drosophila gastrulation, we treated early Drosophila embryos with U73122: this caused a significant (P = 0.0375; Boschloo's test, n=56) delay in the initiation of gastrulation compared to control embryos treated with U73343 (n=34) suggesting that Ca²⁺ activity may also be involved in regulating the initiation of gastrulation in the fly (Extended Data Fig. 8c)."

Is only the onset of gastrulation is delayed? Or other processes also delayed/slower. Normalization is required in order to assess if the effect is on gastrulation onset or on development in general.

We thank the review for this comment. To normalize the results, we have now assessed the extent of cellularisation of the same embryos. We find that the onset of gastrulation, but not the timing of cellularisation, was delayed. We have now added these data to Supplementary Fig. 9c.

Reviewer #3 (Remarks to the Author):

General Comments:

The authors of this paper have employed a combination of computational modeling and experiments to investigate a potential mechanism for the formation of long-range molecular gradients that regulate embryonic polarity during primitive streak formation in early embryogenesis. They observed and quantified transient Ca^{2+} activity spikes, which resulted in signaling waves along the peripheral extraembryonic marginal zone in chick embryos. Through computer simulations, the authors suggest that Ca^{2+} dynamics play a role in regulating embryonic polarity by interacting with the established BMP gradient. While the study is potentially interesting, there are several areas that require improvement, including data analysis, the proposed mathematical model and execution, and comparison between experiments and simulations.

I would like to start by acknowledging the amount of work dedicated to this study and congratulating all the authors for their valuable contribution to this manuscript. The paper is highly readable and engaging, with well-structured figures. The observation and quantification of calcium waves during primitive streak formation are particularly intriguing.

As a reviewer with expertise in mathematical and computational biology, my main concern lies in the mechanistic interpretation of the effect of calcium waves on pre-established morphogenetic signals. The authors have developed a mathematical model to guide the interpretation of the proposed mechanism. Consequently, my review will focus more on this aspect of the manuscript. Several additional steps need to be addressed to ensure that the model comprehensively supports the experimental observations.

1. Complete definition of the model:

- The model equations and boundary conditions require a more thorough explanation. The current presentation lacks both mathematical and descriptive details regarding the boundary conditions.

Thank you for this comment. We have now expanded the explanations throughout.

- The initial conditions should be clearly specified both mathematically and in the text. Additionally, the authors should provide a well-justified rationale for their choice of initial conditions. If the model solutions are sensitive to initial conditions, conducting a scan with different initial conditions would be beneficial. It is acceptable to describe the model in the main text if necessary, but keeping all model details in the methods section would provide greater clarity.

We have now specified the initial conditions in the text. We have now also tested different initial conditions – we have now placed these details in the Supplementary Information so as not to clutter the main text or methods too much.

- Variables and parameters need explicit descriptions, including the provision of units. While the work adequately describes the variables, only a general parameter type description is given in Extended Data Table 2. Providing units for all variables and parameters is essential. Currently, units are only provided for time (in hours) when describing the time discretization step.

Detailed descriptions of the parameters are now included in Supplementary Table 2. We specify the units of distance (mm) and time (hours). For \$\text{Ca}^{2+}\$ activity, however, we have used arbitrary units because “activity” is a complex combination of the amplitude of the spike, its duration and the frequency of firing.

2. Phase space of model solutions:

- To validate the model, it is crucial to explore the phase space of its solutions thoroughly. Conducting

a comprehensive parameter scan will reveal the different solutions that the model can produce. This analysis will help identify parameter ranges that align with experimental observations and provide a set of potential perturbations for experimental validation.

Thank you for these comments. We have now tested a range of parameters, testing where the model fails and in what ways. The results of this are now included in the Supplementary Information (including new tables).

- The authors have only provided one parameter set without justifying their choice. Given the use of parameter-dependent Heaviside functions in the model, variations in the parameter set will strongly influence the model solutions and their fit with experimental observations.

As in the previous question, we have now tested a range of parameter values and also tested the effects of using Hill functions instead of Heaviside. The two upper thresholds (corresponding to the point at which calcium activity initiates BMP expression and at which a low BMP concentration initiates Vg1 expression, respectively) are unaffected by the use of either Heaviside or Hill functions. The strongest threshold, corresponding to where cells “differentiate” to generate a primitive streak is qualitatively different from the other two and reflects an irreversible change – this threshold does require a Heaviside function. However, our exploration of parameter values reveals that this threshold is not the most sensitive to variation. We have now commented on this in the Supplementary Information and added two Supplementary movies (8 and 9).

3. Clarification and modification of the model:

- The transition from calcium waves to a gradient needs to be rationalized, derived, and quantified in terms of gradient formation dynamics, length-scale, time-scale, and the transport mechanism regulating gradient formation. The authors should reference relevant literature on the stability of pulse and traveling waves, such as the fire-diffuse-fire model, to support their proposed hypothesis.

Thank you for this comment. We have now made a number of changes to the manuscript that address this important point. This includes a change of title of the manuscript to emphasize the relationship between short-range interactions (including calcium firing propagation, which we now call “tracks” rather than waves to emphasize their relatively short range) and longer-range consequences like establishment and maintenance of the BMP gradient. There is no long-range gradient in calcium concentration. We also now refer to the relevant literature suggested by this reviewer (see below) referring to the relationship between short and long gradients specifying range positional information. However here the “diffusion” is between subsets of cells in the MZ that are connected to each other by B-type gap junctions rather than diffusion through the extracellular space, and to our knowledge there are no measurements of these parameters in the literature for other epithelial systems at the moment.

- Currently, calcium is modeled as a diffusive molecule using a reaction-diffusion equation that does not allow for solutions in the form of traveling waves. The authors should provide a clear justification for this modeling choice.

First, we realise that our original wording, using “waves” to describe the progress of calcium activity across the embryo was misleading because no wave of calcium concentration can be detected. Rather, the wave is one of calcium activity (in terms of short-range firing trails) that moves across the embryo. To make this clearer, we have removed “calcium waves” from the text and replaced it by “tracks”. In the model, we reflect this by modelling the movement of calcium activity rather than concentration, where “activity” reflects the amplitude, duration and frequency of the calcium pulses. A reaction-diffusion equation is still a valid and simple assumption for the progress of this activity across the embryo.

- Morphogens such as BMP and VG1 are modeled as ordinary differential equations (ODEs) without justification for their homogeneous description.

We model the changes in BMP and Vg1 as functions of their local rates of production and decay, as is standard in the field. We have added a sentence in the text to clarify this.

In conclusion, this work proposes a mechanism for modulating the shape and dynamics of morphogen gradients through the rapid spread of calcium waves. The authors have gathered sufficient experimental data to feed a robust mechanistic model that could strongly support their findings and hypothesis. As a suggestion, the authors could develop a self-consistent model that proposes a mechanism to explain these observations without relying on branched equations. The presented elements and interactions in the model suggest that patterning and the robust establishment of the primitive streak can be self-organized through the interplay of an activator-inhibitor structure and rapid traveling wave spread.

As minor comments:

1. The GitHub link provided in the manuscript does not work. It would be helpful for the authors to review the code and provide further explanation and comments.

Thank you for alerting us to this. This has now been fixed.

2. The authors state that "morphogen gradients can reliably be generated by diffusion only over a limited range (<100 cell diameters, or 1mm)." However, there are examples of robust long-range morphogen gradients, such as in planaria (DOI: 10.1016/j.devcel.2016.12.024, 10.1088/1478-3975/ac86b4). The authors should consider this literature and provide a discussion of how their findings align or differ from those studies.

Thank you for these references. We had missed them. One of them is indeed a proposal for a similar model to the one we are suggesting here, and we have now referred to it. Indeed, our paper supports the theoretical model proposed in that paper.

3. The authors mention that "The diameter of the area pellucida and MZ (excluding the outer extraembryonic area opaca) in pre-primitive-streak stage embryos has an average diameter of 240 cells; therefore, the half-circumference of the MZ would span approximately 380 cell lengths, suggesting that the signal travels faster than 1 cell/min (≥ 1.27 cells/min). This fast transmission rate suggests an intercellular mechanism based on small molecules, such as ion currents." This speed corresponds to an effective diffusion coefficient on the order of $2 \text{ } \mu\text{m}^2/\text{s}$, which can also fit with a diffusion-based morphogen spread.

The reviewer appears to be thinking of simple diffusion occurring through extracellular space. In the epiblast of the chick embryo, the cells are tightly adherent to each other and one-cell-thick, with columnar morphology and almost no extracellular space. Above and below this tissue are very large volumes of fluid which are in motion by turning of the egg and therefore stirred, and unlikely to support the formation of a stable concentration gradient across 3 mm distance. The observed gradients are of gene expression, showing local differences in production of the "morphogens". We have re-focused the paper (including a change of title) to make this more explicit and especially to consider that calcium activity and its progress across the embryo can generate and maintain such gradients without invoking long-range simple diffusion. The transmission of calcium activity between cells through gap junctions only travels a few cell diameters at a time, and involves only a subset of cells in the tissue.

4. The authors mention that "Misexpression of either GJB2 or GJB6 rescued the marginal-zone-excision phenotype, decreasing the frequency of ectopic primitive streak formation relative to both a control plasmid and to misexpression of GJA5 (Fig. 1, d-e), which should not form functional connections with

GJBs (Extended Data Fig. 2, f-g)." If signal spread is a result of morphogen diffusion, altering junctions can change the transport dynamics of such a morphogen, resulting in slower gradient formation and a shorter-range length-scale. Have the authors extended the timescale of this experiment to observe delayed patterning? Additionally, short-range gradients resulting from altered junctions can potentially explain changes in the likelihood of primitive streak formation.

We have not changed the timescale to study the timing of streak formation, but have grown the embryos more than long enough to allow for the formation of a mature primitive streak. Doing this in time-course would require hundreds of embryos and would be a huge undertaking. We did not feel that observing a delay (which was not suggested by cursory observations during incubation) in some of the conditions would particularly affect the conclusions from this experiment. The main findings relate to the differences in activity between A- and B-type gap junctions in terms of rescuing the flow of positional information around the excisions.

5. The authors mention that "We explored this further using mosaic electroporation of an expression vector encoding the Ca²⁺-reporter GCaMP6; high-resolution imaging revealed very thin intercellular processes connecting firing cells that were not adjacent (Fig. 2 bi-biv and Extended Data Movie 4). These processes are reminiscent of the cytonemes described in *Drosophila*¹⁹ and other systems, including the somite in early chick embryos²⁰. These observations suggest that Ca²⁺ spikes may be transmitted via cytoneme-like processes (Extended Data Movie 4)." However, the notion of signal transport via cytonemes is still controversial and lacks clear agreement in *Drosophila*. Evidence for signal transport via cytonemes in chick embryos is limited, and the authors need to conduct a thorough study to support this claim. I suggest that the authors either provide stronger evidence for this type of transport mechanism or temper their language and suggest complementary and alternative transport possibilities.

We have purposely remained circumspect in these statements. We do not say that the observed processes ARE cytonemes, but only that the processes we observe resemble what has been described in *Drosophila* and a few other systems. The transmission of the calcium signal through these processes is seen clearly in the movie clip included and we simply describe what we have seen. We are not drawing conclusions beyond this and do not use it in the remainder of the paper.

6. The authors state that "The above results suggest that relatively short-range transmission of Ca²⁺ transients conveys positional information over a large embryonic territory to regulate the site of primitive streak formation." It would be helpful to clarify the length of the range referred to in this statement. Previously, it was mentioned that Ca²⁺ waves can span 200 μ m, which is not considered short-range in the context of information spread. It may be beneficial to provide this distance in the context of other long-range signals in this system.

There have been several changes made to the manuscript to address this and be more explicit. We hope that the scenario we propose is now clearer. As mentioned above, we are not proposing "waves of calcium" travelling across large distances but rather short tracks and have changed the terminology accordingly.

7. When the authors propose that "we hypothesized that a gradient of Ca²⁺ activity, rather than Ca²⁺ concentration, along the MZ might act as the positional cue," please clarify the difference between Ca²⁺ activity and Ca²⁺ concentration. Additionally, as mentioned earlier, this hypothesis requires more background information. The authors have described the observation of traveling Ca²⁺ waves, but transitioning from traveling waves to stable gradient formation necessitates further reasoning before proposing this hypothesis.

This is the same point we address above and in response to other reviewers. The changes we have

made to the manuscript including the change of title should address it directly. We agree with the reviewer that this was misleading in the original version.

Overall, the paper has great potential; however, improvements are necessary in the mathematical model, and the mechanistic interpretation to support the conclusions. I encourage the authors to address the mentioned concerns for a more robust and comprehensive understanding of the experimental observations.

Dr. Daniel Aguilar-Hidalgo

REVIEWERS' COMMENTS

Reviewer #1 (Remarks to the Author):

The authors have made a serious attempt to answer my main queries and comments. The added experiments/modeling and clarifications provided have improved the paper. Although not all points raised have been completely resolved, the authors have given reasonable explanations why addressing some is difficult at present (local manipulation of calcium levels as well as embryo scale imaging of calcium dynamics) it is my opinion that there are definitively enough novel and interesting observations and hypotheses, that merit publication in its present form

Reviewer #2 (Remarks to the Author):

The manuscript by Lee et al. proposes an interesting interplay between BMP signaling and calcium activity for primitive streak positioning. While I think that several questions remain in regards to the calcium waves/activity spikes and how they interact with BMP signaling, in part due to technical limitations, the authors do now present more and more convincing evidence and/or adjusted the text where appropriate to substantiate their key conclusions in this manuscript. Given the interest in the process and its underlying mechanism, I think that this manuscript can serve as a critical starting point for more in-depth analysis of the role of Calcium signaling together with morphogen signaling for primitive streak formation.

Reviewer #3 (Remarks to the Author):

I would like to thank the authors for their efforts in addressing all the raised points during the review process. I am very happy to see the new shape of this great work.

At this point, I only have one concern. This is related to the parameter scan in the model. First, thanks for the performed scanned and for the detailed table explaining effects on parameter value modification.

My concern is that for most parameters, the values has been modified in a 0.1-fold from their original value. In many cases, this small variation would land within experimental error. There are two consequences to this:

1. Such a narrow span prevents finding other potential solutions in the model, which could validate perturbations made to the system.
2. Makes it difficult to distinguish between sensitive and insensitive parameters to explain experimental results. Parameters that show as insensitive may seem so as the change is too small. Actually, some of the parameters seem to be very sensitive and provide solutions not matching experimental observations, which lead to my next comment.

Most importantly, as mentioned above, a 0.1-fold from the original value can land within experimental parameter deviation. The way the model is presented right now can lead to the interpretation that the model is valid to explain observations just for one parameter value set, which highly decreases the value of the model itself.

I encourage the authors to increase the parameter range in the scan such that it is clear that no other solutions/trends will be found. Additionally, the authors should provide a parameter range where the model can qualitatively explain experimental observations. Please make sure no parameter is so sensitive that the model can only explain experimental observations for a very narrow parameter range. This may imply that the model is ill-defined.

A quantitative fit would be great, but I will leave this as optional as I did not mentioned this in my

previous revision, and the authors has referred to the model as a qualitative model.

Dr. Daniel Aguilar-Hidalgo

Dear editor and reviewers:

RE: Nature Communications manuscript NCOMMS-23-24093B (re-revised)

We are very pleased by the latest email about this manuscript, with good news about the likely acceptance for publication. We have followed all editorial guidelines including all of those in the “Editorial Requests”, completed new versions of the Reporting Summary and Editorial Policy checklist, shortened the abstract to 150 words (with no references), the subtitles to less than 60 characters, subdivided some of the main and supplementary figures so that all legends became shorter than 350 words each, and made a number of other changes throughout to comply with all the requests and stylistic requirements. There was one remaining minor comment from Reviewer 3. The reviewer’s key point is:

I encourage the authors to increase the parameter range in the scan such that it is clear that no other solutions/trends will be found. Additionally, the authors should provide a parameter range where the model can qualitatively explain experimental observations. Please make sure no parameter is so sensitive that the model can only explain experimental observations for a very narrow parameter range. This may imply that the model is ill-defined. A quantitative fit would be great, but I will leave this as optional as I did not mention this in my previous revision, and the authors has referred to the model as a qualitative model.

We have decided to follow this request in full. For all parameters where the model previously “failed”, we have estimated the FULL RANGE within which the model performs like the biological system and the results are shown in Supplementary Table 3 in the Supplementary Information file. All ranges are quite broad so no single parameter is disproportionately sensitive. We explain that when a parameter value causes the model to fail, it does so in one of two ways, both of which make intuitive biological sense. We also end the Supplementary Information section on this topic with a comment about “Regulative systems”, explaining that in such systems, it is expected that real embryos would deploy a compensatory mechanism to balance for variation in various real parameters and still achieve appropriate outcomes, therefore parameter estimations like these are likely to be a particularly stringent test of robustness for the model.

We are confident that the paper has improved very significantly as a result of these rounds of review and as before, we greatly appreciate the professional attitude and helpfulness of the reviewers and the editor in helping us to improve this paper.